# SeWA: Selective Weight Average via Probabilistic Masking

## Abstract

Weight averaging has become a standard technique for enhancing model performance. However, methods such as Stochastic Weight Averaging (SWA) and Latest Weight Averaging (LAWA) rely on manually designed checkpoint selection rules, which struggle under unstable training dynamics. To minimize human bias, this paper proposes Selective Weight Averaging (SeWA), which adaptively selects checkpoints during the final stages of training for averaging. Both theoretically and empirically, we show that SeWA achieves a better generalization. From an algorithm implementation perspective, SeWA can be formulated as a discrete subset selection problem, which is inherently challenging to solve. To address this, we transform it into a continuous probabilistic optimization framework and employ the Gumbel-Softmax estimator to learn the non-differentiable mask for each checkpoint. Theoretically, we first prove that SeWA converges to a critical point with flatter curvature, thereby explaining its underlying mechanism. We further derive stability-based generalization bounds for SeWA, which are sharper than those of SGD under both convex and non-convex assumptions, thus providing formal guarantees of improved generalization. Finally, extensive empirical evaluations across diverse domains, including behavior cloning, image classification, and text classification, demonstrate the robustness and effectiveness of our approach.

## 1 Introduction

Model averaging has shown substantial benefits in deep learning, both in empirical performance across practical applications and in theoretical analyses related to generalization and optimization. From the perspective of generalization, averaging-based algorithms, such as SWA Izmailov et al. (2018), Exponential Moving Average (EMA) Szegedy et al. (2016), LAWA (Kaddour, 2022; Sanyal et al., 2023), and Trainable Weight Averaging (TWA) (Li et al., 2022), have been empirically validated to enhance generalization performance across various tasks. These methods have gained widespread adoption in several domains, including large-scale network training (Izmailov et al., 2018; Lu et al., 2022; Sanyal et al., 2023) and adversarial learning (Xiao et al., 2022). In theoretical research, Hardt et al. (2016) and Xiao et al. (2022) successively give stability-based generalization bounds for SWA in different application contexts, showing that under the convexity assumption, the generalization bound of the SWA algorithm is half that of SGD. From an optimization perspective, model averaging can facilitate convergence by stabilizing the trajectory of the optimizer when it oscillates near a local minimum. Polyak & Juditsky (1992) demonstrate that averaging model weights improves convergence speed in the setting of convex loss functions. More recently, Sanyal et al. (2023) have empirically verified accelerated convergence using the LAWA in Large Language Models pre-training.

Despite their theoretical and empirical advantages, averaging-based algorithms often depend on manually designed training frameworks and are sensitive to hyperparameter selection. For example, SWA revisits historical model states at each step, which can slow convergence, and requires a cyclic learning rate schedule to identify low-loss regions, introducing additional tuning overhead. In contrast, LAWA selects the final averaging point from the last $k$ epochs. However, Sanyal et al. (2023) have observed that performance does not vary monotonically with respect to $k$; instead, it improves initially and then degrades as $k$ increases. TWA addresses some of these limitations by adaptively learning averaging weights, but it incurs extra computational cost due to the need for orthogonalizing two subspaces.

In this paper, we propose a novel averaging algorithm that minimizes the reliance on manually designed training frameworks while balancing generalization and training stability. SeWA adaptively learns aggregation weights from the last $k$ steps of the SGD training trajectory, thereby mitigating the influence of early-stage information and enhancing the performance gains achieved through model averaging. This adaptive integration mechanism not only reduces the need for

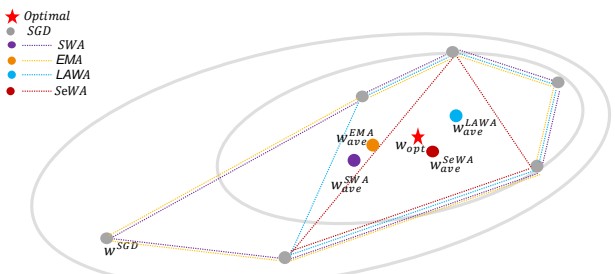

Figure 1: Comparison of SeWA with different models on convergence performance.

extensive hyperparameter tuning but also mitigates performance degradation caused by redundant or suboptimal weight selections. As shown in Figure 1, SeWA achieves near-optimal performance, achieving flatter minima compared to existing approaches.

During the implementation and theoretical analysis of our algorithm, we encountered three key challenges: (1) The adaptive selection of checkpoints can be formulated as a subset selection task, a typical discrete optimization problem. Solving such problems requires handling discrete variables that are often non-differentiable. (2) Establishing the stability-based generalization bound for SeWA requires not only quantifying the impact of input perturbations on the output but also analyzing the influence introduced by the adaptive learning process. (3) Although stability-based generalization bounds provide theoretical guarantees of desirable properties, they do not explain the intrinsic operational mechanisms of SeWA, leaving its functioning essentially a black box.

To address these challenges, we formulate the SeWA solving process as the coreset selection problem, embedding the discrete optimization objective into a probabilistic space, which enables the utilization of gradient-based continuous optimization methods. Furthermore, we employ the Gumbel-softmax estimator to address the non-differentiability of binary variables. In generalization analysis, the discrete selection problem of adaptive learning processes is transformed, in expectation, into a global averaging process dependent on selection probabilities, establishing a theoretical bridge for building SeWA's stability bounds. We also derive generalization bounds for SeWA under different assumptions based on stability, which are sharper than those of other algorithms (see Table 1). Furthermore, based on the differential form of the derivative of our relaxation function, we establish that SeWA converges to a critical point with flatter landscape. Finally, extensive experiments have been conducted across various domains, including computer vision, natural language processing, and reinforcement learning, confirming the algorithm's generalization advantages. Our contributions are listed as follows.

- Our approach adaptively selects models for averaging in the final training stages, ensuring strong generalization, lower manual cost, and reduced bias toward specific scenarios. Notably, the selection paradigm of SeWA is well-suited to unstable training processes (e.g., reinforcement learning), where it yields significant improvements in generalization.

- We propose a solvable optimization framework by transforming the discrete problem into a continuous probabilistic space and addressing the non-differentiability of binary variables using the Gumbel-Softmax estimator during optimization.

- We prove that the SeWA can converge to a critical point with flatter curvature, thereby providing a theoretical foundation for understanding its underlying mechanism. Further, we analyze the impact of masks on generalization theory in expectation and derive a stability-based generalization upper bound for SeWA, showing advantages over SGD and other averaging-based algorithms' bounds under the different function assumptions.

- We empirically demonstrate the outstanding performance of our algorithm in multiple domains, including behavior cloning, image classification, and text classification. In particular, the SeWA achieves comparable performance using only a few selected points, matching or exceeding the performance of other methods that require many times more points.

**Related Work**, Due to space limitations, the comprehensive literature review is placed in Appendix A. In particular, we present a detailed comparison of the generalization bounds for our proposed SeWA and existing algorithms in Table 1.

Table 1: Comparison of SeWA with other algorithms on different settings. Here $T$ represents iterations, and $n$ denotes the size of the datasets. $L$, $\beta$, and $c$ are constants. $k$ is the number of averages. $\hat{s} = \sup_{T-k+1 \leq i \leq T} s_i$, $\hat{s} \in (0,1]$, $s_i$ corresponds to the probability of mask $m = 1$ and $\mathcal{O}_{\hat{s}}$ means that this upper bound depends on $\hat{s}$. We can derive that SeWA has sharper bounds compared to others in different settings, where FWA is the general form of LAWA.

| SETTINGS | LEARNING RATE | ALGORITHM | GENERALIZATION BOUND |
|---|---|---|---|
| CONVEX | $\alpha_t = \alpha$ | SGD | $2\alpha LT/n$ HARDT ET AL. (2016) |
| | | SWA | $\alpha LT/n$ XIAO ET AL. (2022) |
| | | FWA | $2\alpha L(T - k/2)/n$ WANG ET AL. (2024B) |
| | | EMA | $-$ |
| | | SEWA | $2\alpha L\hat{s}(T - k/2)/n$ THEOREM 4.6 |
| NON-CONVEX | $\alpha_t = \frac{c}{t}$ | SGD | $\mathcal{O}(T^{\frac{c\beta}{1+c\beta}}/n)$ HARDT ET AL. (2016) |
| | | SWA | $\mathcal{O}(T^{\frac{c\beta}{2+c\beta}}/n)$ WANG ET AL. (2024A) |
| | | FWA | $\mathcal{O}(T^{\frac{c\beta}{k+c\beta}}/n)$ WANG ET AL. (2024B) |
| | | EMA | $-$ |
| | | SEWA | $\mathcal{O}_{\hat{s}}(T^{\frac{c\beta}{k+c\beta}}/n)$ THEOREM 4.11 |

## 2 METHODOLOGY

In this section, we begin by formalizing the problem setup and introducing the foundational assumptions, definitions, and key properties. We then present the proposed SeWA algorithm along with the essential terminology required for its understanding.

### 2.1 PROBLEM SETTING

Let $F(w, z)$ be a loss function that measures the loss of the predicted value of the network parameter $w$ at a given sample $z$. There is an unknown distribution $\mathcal{D}$ and a sample dataset $S = (z_1, z_2, ..., z_n)$ of $n$ examples i.i.d. drawn from $\mathcal{D}$. Then the *population risk* and *empirical risk* are defined as

$$\textbf{Population Risk:} R_{\mathcal{D}}[w] = E_{z \sim \mathcal{D}} F(w; z) \quad \textbf{and} \quad \textbf{Empirical Risk:} R_S[w] = \frac{1}{n} \sum_{i=1}^{n} F(w; z_i).$$

The generalization error of a model $w$ is the difference $\epsilon_{gen} = R_{\mathcal{D}}[w] - R_S[w]$.

**SGD**. For the target function $F$ and the given dataset $S = (z_1, z_2, \cdots, z_n)$, we consider the SGD's general update rule as

$$w_{t+1} = w_t - \alpha \nabla_w F(w_t, z_{i_t}), \tag{1}$$

where $\alpha$ is the fixed learning rate, $z_{i_t}$ is the sample chosen in iteration $t$. We choose $z_{i_t}$ from dataset $S$ in a standard way, picking $i_t \sim \text{Uniform}\{1, \cdots, n\}$ at each step. This setting is commonly explored in analyzing the stability Hardt et al. (2016); Xiao et al. (2022).

**SeWA** algorithm adaptively selects $K$ points for averaging among the last $k$ points on the training trajectory after $T$ steps of the SGD iterations. It is formulated as

$$\bar{w}_T^K = \frac{1}{K} \sum_{i=T-k+1}^{T} m_i w_i, \tag{2}$$

where the mask $m_i \in \{0, 1\}$ and $m_i = 1$ indicating the $i$-th weight is selected for averaging and otherwise excluded; the selection count $K = k_{m_i=1} = \sum_{i=T-k+1}^{T} m_i$ quantifies the number of selected weights within the interval $[T - k + 1, T]$, which equivalently represents the number of candidate models incorporated in averaging. In practice, the SeWA algorithm selects the top-K highest-probability weights for averaging, as outlined in Algorithm 1.

---

**Algorithm 1:** Selective Weight Average

**Input:** Checkpoints $\mathbf{w}$, hyper-parameters $t, M, max\_iteration$

**Init:** Mask probability $s$;

1 **for** $i = 1, \ldots, max\_iteration$ **do**
2    Gumbel-softmax sampling **for** $m = 1, \ldots, M$ **do**
3      Sample $u^{(m)} \sim \text{Uniform}(0,1)$;
4      Compute $F\left(\mathbf{w}(\text{GS}(s, u^{(m)}, t)\right)$;
5    **end**
6    Learning mask probability Optimize $\hat{F}(s) = \frac{1}{M} \sum_{m=1}^{M} F\left(\mathbf{w}(\text{GS}(s, u^{(m)}, t)\right)$;
7 **end**

**Output:** Mask $m$ based on $K$ largest probabilities in $s$

---

## 2.2 Basic Assumptions

Moreover, we assume function $F$ satisfies the following *Lipschitz* and *smoothness* assumption.

**Assumption 2.1** ($L$-Lipschitz). A differentiable function $F : R^d \to R$ satisfies the $L$-Lipschitz property, i.e., for $\forall u, v \in R^d, \|F(u) - F(v)\| \leq L\|u - v\|$, which implies $\|\nabla F(u)\| \leq L$.

**Assumption 2.2** ($\beta$-smooth). A differentiable function $F : R^d \to R$ is $\beta$-smooth, i.e., for $\forall u, v \in R^d$, we have $\|\nabla F(u) - \nabla F(v)\| \leq \beta\|u - v\|$.

Assumptions 2.1 and 2.2 are often used to establish stability bounds for algorithms and are crucial conditions for analyzing the model's generalization performance.

**Assumption 2.3** (Convex function). A differentiable function $F : R^d \to R$ is convex, i.e., for $\forall u, v \in R^d, F(u) \leq F(v) + \langle \nabla F(u), u - v \rangle$.

Different functional assumptions correspond to different expansion properties, which determine the different generalization bounds and will be discussed in Lemma 2.4 and Chapter 4.

## 2.3 The Expansive Properties

**Lemma 2.4.** *Assume that the function $F$ is $\beta$-smooth. Then,*
*(1). (non-expansive) If $F$ is convex, for any $\alpha \leq \frac{2}{\beta}$, we have $\|w_{T+1} - w'_{T+1}\| \leq \|w_T - w'_T\|$;*
*(2). (($1+\alpha\beta$)-expansive) If $F$ is non-convex, for any $\alpha$, we have $\|w_{T+1} - w'_{T+1}\| \leq (1+\alpha\beta)\|w_T - w'_T\|$.*

Lemma 2.4 tells us that the gradient update becomes $non$-expansive when the function is convex and the step size is small, which implies that the algorithm will always converge to the optimum in this setting. However, although this is not guaranteed when the function is non-convex, it is required that the gradient updates cannot be overly expansive if the algorithm is stable. The proof of Lemma 2.4 is deferred to Appendix C. Additional dissuasion can be found in Hardt et al. (2016); Xiao et al. (2022).

## 2.4 Stability and Generalization Definition

Hardt et al. (2016) link the *uniform stability* of the learning algorithm with the expected generalization error bound in research of SGD's generalization. The expected generalization error of a model $w = A_S$ trained by a certain randomized algorithm $A$ is defined as

$$\mathbb{E}_{S,A}\left[R_S\left[A_S\right] - R_{\mathcal{D}}\left[A_S\right]\right]. \tag{3}$$

Here, expectation is taken over the internal randomness of $A$. Next, we introduce the *uniform stability*.

**Definition 2.5** ($\epsilon$-Uniformly Stable). A randomized algorithm $A$ is $\epsilon$-uniformly stable if for all data sets $S, S'$ from $\mathcal{D}$ such that $S$ and $S'$ differ in at most one example, we have

$$\sup_{z \in S, S'} \left\{\mathbb{E}_A\left[F(A_S; z) - F(A_{S'}; z)\right]\right\} \leq \epsilon. \tag{4}$$

**Theorem 2.6.** (Generalization in Expectation (Hardt et al., 2016, Theorem 2.2)) *Let $A$ be $\epsilon$-uniformly stable. Then,*

$$\left|\mathbb{E}_{S,A}\left[R_S\left[A_S\right] - R_{\mathcal{D}}\left[A_S\right]\right]\right| \leq \epsilon. \tag{5}$$

This theorem clearly states that if an algorithm has uniform stability, then its generalization error is small. In other words, uniform stability implies *generalization in expectation* Hardt et al. (2016). Above proof is based on Bousquet & Elisseeff (2002, Lemma 7) and similar to Shalev-Shwartz et al. (2010, Lemma 11).

## 3 Practical SeWA Implementation

Although the SeWA algorithm has simpler expressions, the difficulty is learning the mask $m_i$. Inspired by tasks such as coreset selection Zhou et al. (2022), the discrete problem is relaxed to a continuous one. We first formulate weight selection into the following discrete optimization paradigm:

$$\min_{m \in C} F(m) = F\left(\mathbf{w}(m)\right) = \frac{1}{n}\sum_{i=1}^{n} F\left(\mathbf{w}(m); z\right), \tag{6}$$

where $C = \{\mathbf{m} : m_i = 0 \text{ or } 1, \|\mathbf{m}\|_0 \le K\}$ and $\mathbf{w}(m) = \frac{1}{K} \sum_{i=T-k+1}^{T} m_i w_i$.

To transform the discrete Eq. 6 into a continuous one, we treat each mask $m_i$ as an independent binary random variable and reparameterize it as a Bernoulli random variable, $m_i \sim \text{Bern}(s_i)$, where $s_i \in [0, 1]$ represents the probability of $m_i$ taking the value 1, while $1 - s_i$ corresponds to the probability of $m_i$ being 0. Consequently, the joint probability distribution of $m$ is expressed as $p(m|s) = \prod_{i=1}^{n} (s_i)^{m_i} (1 - s_i)^{1-m_i}$. Then, the feasible domain of the target Eq. 6 approximately becomes $\hat{C} = \{s : 0 \le s \le 1, \|s\|_1 \le K\}$ since $\mathbb{E}_{m_i \sim p(m|s)} \|m\|_0 = \sum_{i=1}^{n} s_i$. As in the previous definition, $K > 0$ in $\hat{C}$ is a constant that controls the size of the feasible domain. Then, Eq. 6 can be naturally relaxed into the following excepted loss minimization problem:

$$\min_{s \in \hat{C}} F(s) = \mathbf{E}_{p(m|s)} F(\mathbf{w}(m)), \tag{7}$$

where $\hat{C} = \{s : 0 \le s \le 1, \|s\|_1 \le K\}$. Optimizing Eq.7 involves discrete random variables, which are non-differentiable. One choice is using Policy Gradient Estimators (PGE) such as the REIN-FORCE algorithm (Williams, 1992; Sutton et al., 1999) to bypass the back-propagation of discrete masks $m$,

$$\nabla_s F(s) = \mathbf{E}_{p(m|s)} F(\mathbf{w}(m)) \nabla_s \log p(m \mid s).$$

However, these algorithms suffer from the high variance of computing the expectation of the objective function, hence may lead to slow convergence or sub-optimal results.

To overcome these issues, we resort to the reparameterization trick using Gumbel-softmax sampling (Jang et al., 2017; Maddison et al., 2017). Instead of sampling discrete masks $m$, we get continuous relaxations by,

$$\tilde{m}_i = \frac{\exp((\log s_i + g_{i,1})/t)}{\exp((\log s_i + g_{i,1})/t) + \exp((\log(1 - s_i) + g_{i,0})/t)}, \tag{8}$$

for $i = 1, \ldots, k$, where $g_{i,0}$ and $g_{i,1}$ are i.i.d. samples from the Gumbel$(0, 1)$ distribution. The hyperparameter $t > 0$ controls the sharpness of this approximation. When it reaches zero, i.e., $t \to 0$, $\tilde{m}$ converges to the true binary mask $m$. During training, we maintain $t > 0$ to ensure the function is continuous. For inference, we can sample from the Bernoulli distribution with probability $s$ to get sparse binary masks. In practice, the random variables $g \sim \text{Gumbel}(0, 1)$ can be sampled from,

$$g = -\log(-\log(u)), \quad u \sim \text{Uniform}(0, 1).$$

For simplicity, we denote the Gumbel-softmax sampling in Eq. 8 as $\tilde{m} = \text{GS}(s, u, t)$, where $u \sim \text{Uniform}(0, 1)$. Replacing the binary mask $m$ in Eq. 7 with the continuous relaxation $\tilde{m}$, the optimization problem becomes,

$$\min_{s \in \hat{C}} F(s) = \mathbf{E}_{u \sim \text{Uniform}(0, 1)} F(\mathbf{w}(\text{GS}(s, u, t)), \text{ where } \hat{C} = \{s : 0 \le s \le 1, \|s\|_1 \le K\}.$$

The expectation can be approximated by Monte Carlo samples, i.e.,

$$\min_{s \in \hat{C}} \hat{F}(s) = \frac{1}{M} \sum_{m=1}^{M} F\left(\mathbf{w}(\text{GS}(s, u^{(m)}, t)\right),$$

where $u^{(m)}$ are i.i.d. samples drawn from Uniform$(0, 1)$. Empirically, since the distribution of $u$ is fixed, this Monte Carlo approximation exhibits low variance and stable training Kingma & Welling (2013); Rezende et al. (2014). Furthermore, since Eq. 8 is continuous, we can optimize it using back-propagation and gradient methods.

*Remark* 3.1. SeWA adaptively selects useful checkpoints, which implies that it does not require the extra cost associated with manual design and avoids model biases introduced by prior knowledge, thereby making our approach applicable to a broader range of tasks. In the following experiments, SeWA algorithm demonstrates particular suitability for scenarios characterized by unstable training trajectories, such as behavior cloning. By leveraging checkpoint averaging, SeWA effectively stabilizes the training process, mitigating fluctuations and enhancing overall performance.

# 4 THEORETICAL ANALYSIS OF SEWA

## 4.1 OPTIMIZATION ANALYSIS

Next, we show that the standard gradient descent algorithm will converge to a "flat" point. Prior to this, we first revisit the definition of stationary points for the minimization problem, that is to say,

**Definition 4.1.** Given a differentiable function $G : \mathcal{K} \to \mathbb{R}$ and a domain $\mathcal{C} \subseteq \mathcal{K}$, a point $\mathbf{x} \in \mathcal{C}$ is called as a stationary point for the function $G$ over $\mathcal{C}$ if and only if $\min_{\mathbf{y} \in \mathcal{C}} \langle \mathbf{y} - \mathbf{x}, \nabla G(\mathbf{x}) \rangle \geq 0$.

Then, we have the following result

**Theorem 4.2.** *If the Bernoulli extension $F$ in Eq.7 is $\beta$-smooth, gradient descent with a step size smaller than $\frac{1}{\beta}$ will eventually converge to a stationary point.*

*Remark* 4.3. The $\beta$-smoothness of $F(s)$ has been verified in Appendix C of (Hassani et al., 2017).

From the definition of Bernoulli extension $F(s)$, we can show that (Calinescu et al., 2011)

$$\frac{\partial F}{\partial s_i}(s) \triangleq \mathbf{E}_{p(m|s)} \Big( F\left(\mathbf{w}(m; m_i \to 1)\right) - F\left(\mathbf{w}(m; m_i \to 0)\right) \Big), \tag{9}$$

where $s \triangleq (s_1, \ldots, s_n) \in [0, 1]^n$, $(m; m_i \to 1)$ means that we reset the $i$-th coordinate of $m$ to 1 and $(m; m_i \to 0)$ denotes setting $m_i$ to value 0.

According to Eq.9, we can infer that $\frac{\partial F}{\partial s_i}(s)$ corresponds to the expected marginal effect of the $i$-th SGD iteration on mask $m$. Generally speaking, gradient descent algorithm only can be constrained to a finite number of iterations. Consequently, the outcome $s$ we finally obtain is an approximate stationary point for Bernoulli extension $F$ with $|\langle y - s, \nabla F(s) \rangle| \leq \epsilon, \forall y \in C$. Particularly, when $s$ is an interior point (near the boundary) of $C$, we can know, for any basic vector $\mathbf{e}_i$, there exists a constant $\lambda$ such that $s \pm \lambda \mathbf{e}_i \in C$, which implies that the following inequality holds:

$$|\lambda \cdot \mathbf{E}_{p(m|s)} \Big( F\left(\mathbf{w}(m; m_i \to 1)\right) - F\left(\mathbf{w}(m; m_i \to 0)\right) \Big)|$$
$$= \max \Big( \langle (s + \lambda \mathbf{e}_i) - s, \nabla F(s) \rangle, \langle (s - \lambda \mathbf{e}_i) - s, \nabla F(s) \rangle \Big) \leq \max_{y \in C} |\langle y - s, \nabla F(s) \rangle| \leq \epsilon. \tag{10}$$

Eq.10 implies that the expected *marginal change of $F(\mathbf{w}(m))$ along any coordinate* is bounded by $\frac{\epsilon}{\lambda}$. In other words, the SeWA algorithm can converge to a critical point with *flatter* curvature.

## 4.2 GENERALIZATION ANALYSIS

This section provides the upper bounds on generalization in the convex and non-convex settings, respectively. First, a critical lemma is provided for building a stability bound in the convex setting.

**Lemma 4.4.** *Let $\bar{w}_T^K$ and $\bar{w}_T^{K'}$ denote the corresponding outputs of SeWA after SGD running $T$ steps on the datasets $S$ and $S'$, which have $n$ samples but only one different. Assume that function $F(\cdot, z)$ satisfies Assumptions 2.1 for a fixed example $z$, then we have*

$$\mathbb{E}|F(\bar{w}_T^K; z) - F(\bar{w}_T^{K'}; z)| \leq \hat{s} L \mathbb{E}[\bar{\delta}_T], \tag{11}$$

*where $\bar{\delta}_T = \frac{1}{k} \sum_{i=T-k+1}^{T} \|w_i - w_i'\|$, $w_i$ and $w_i'$ are the outputs of SGD, and $\hat{s} = \sup_{T-k+1 \leq i \leq T} s_i$, where $s_i$ is the probability of $m_i = 1$ and $\hat{s} \in (0, 1]$.*

*Remark* 4.5. The parameter $\hat{s}$ is the upper bound of the probability $s_i$ that selects a candidate model $w_i$ for averaging. Notably, setting $\hat{s} \neq 0$ carries practical significance: if $\hat{s} = 0$, the algorithm would result in the failure to select any weights for averaging, thereby collapsing model parameters to zero. Such a scenario is incompatible with the algorithm's design principles and fundamentally undermines its intended purpose. Additionally, since the learned probability $s_i$ is inherently encoded within the network parameters, $\hat{s} = 0$ would force all parameters to zero, violating the algorithm's operational framework. Thus, our $\hat{s} \in (0, 1]$ setting is theoretically and practically justified.

The Lemma 4.4 further decomposes the problem of selecting points for averaging within the last $k$ steps into averaging over the last $k$ steps multiplied by the probability $s_i$ of each step by taking an expectation over the mask, which makes it possible further to establish SeWA's stability bounds. Next, we give the bound for SeWA in the convex setting combined with Lemma 4.4.

**Theorem 4.6.** *Suppose that we first run SGD with constant step sizes $\alpha \leq \frac{2}{\beta}$ for $T$ steps, where each step samples $z$ uniformly with replacement and learn the probability $s_i$ of each weight $w_i$ from $k$ checkpoints. If function $F$ satisfies Assumptions 2.1, 2.2 and 2.3. SeWA has uniform stability of*

$$\epsilon_{gen} \leq \frac{2\alpha L^2 \hat{s}}{n} \left(T - \frac{k}{2}\right), \tag{12}$$

*where $\hat{s} = \sup_{T-k+1 \leq i \leq T} s_i$ and $\hat{s} \in (0, 1]$.*

*Remark* 4.7. Theorem 4.6 shows that the SeWA algorithm has a sharper stability bound of $2\alpha L^2 (T - k/2) \hat{s}/n$ under the convex assumption than the bound $2\alpha L^2 T/n$ for SGD given by Hardt et al. (2016). The reason for improving the generalization comes from two main sources: (1) the last $k$ checkpoints averaging improves the SGD bound $\mathcal{O}(T/n)$ to $\mathcal{O}((T - k/2)/n)$. This result degenerates to the SGD bound when $k = 1$. (2) The algorithm further improves the stability bound $2\alpha L^2(T - k/2)/n$ to $\hat{s}$ times its size, which reflects the influence of selection on the bound.

*Remark* 4.8. The $k$ in Theorem 4.6 implies that the more checkpoints involved in the averaging, the better the generalization performance. In practice, $k$ is set sufficiently large to ensure that the selected checkpoints can comprehensively explore the solution space. In contrast, a small $k$ leads to limited improvement in generalization due to the similar performance of checkpoints collected in later stages.

*Remark* 4.9. Theorem 4.6 introduces a scaling parameter $\hat{s}$, which is confined to $(0, 1]$ and linearly modulates the bound $2\alpha L^2(T - k/2)/n$ but remains independent of the number of selected weights. Furthermore, our empirical analysis in Section 5 demonstrates that smaller numbers of selected weights do not consistently yield better generalization performance.

**Lemma 4.10.** *Let $\bar{w}_T^K$ and $\bar{w}_T^{K\prime}$ denote the corresponding outputs of SeWA after SGD running $T$ steps on the datasets $S$ and $S'$, which have $n$ samples but only one different. Assume that function $F(\cdot, z)$ satisfies Assumption 2.1 for a fixed example $z$ and every $t_0 \in \{1, \cdots, n\}$, then we have*

$$\mathbb{E}|F(\bar{w}_T^K; z) - F(\bar{w}_T^{K\prime}; z)| \leq \frac{t_0}{n} + \hat{s}L\mathbb{E}\left[\bar{\delta}_T | \bar{\delta}_{t_0} = 0\right], \tag{13}$$

*where $\bar{\delta}_T = \frac{1}{k}\sum_{i=T-k+1}^{T} \|w_i - w_i'\|$, $w_i$ and $w_i'$ are the outputs of SGD, and $\hat{s} = \sup_{T-k+1 \leq i \leq T} s_i$, where $s_i$ is the probability of $m_i = 1$ and $\hat{s} \in (0, 1]$.*

**Theorem 4.11.** *Suppose we first run SGD with decay step sizes $\alpha \leq \frac{c}{t}$ for $T$ steps, where each step samples $z$ uniformly with replacement and learn the probability $s_i$ of each weight $w_i$ from $k$ checkpoints. Let function $F \in [0, 1]$ satisfies Assumptions 2.1 and 2.2. SeWA has uniform stability of*

$$\epsilon_{gen} \leq \mathcal{O}_{\hat{s}}\left(\frac{T^{\frac{c\beta}{k+c\beta}}}{n}\right), \tag{14}$$

*where $\hat{s} = \sup_{T-k+1 \leq i \leq T} s_i$, $\hat{s} \in (0, 1]$, and $c > 0$ is a constant.*

*Remark* 4.12. In non-convex setting, Theorem 4.11 shows that SeWA has bound $\mathcal{O}(T^{c\beta/(c\beta+k)}/n)$ compared to the $\mathcal{O}(T^{c\beta/(c\beta+1)}/n)$ for SGD in Hardt et al. (2016), showing its ability to improve generalization significantly. Although the number $k$, closely related to the iterations $T$, seems to dominate the result, the direct influence of parameter $\hat{s}$ on the entire bound also plays a crucial role.

*Remark* 4.13. The assumption that $F(w; z) \in [0, 1]$ in Theorem 4.11 is adopted for simplicity. Removing this condition does not affect the final results, as it merely introduces a constant scaling factor. The same setting is commonly used and discussed in Hardt et al. (2016); Xiao et al. (2022).

*Remark* 4.14. We derive the generalization bound of SeWA via stability analysis, following a standard pipeline. As part of this, we establish the bound for averaging the last $k$ iterates, similar to the paper Wang et al. (2024b), but with two key differences: (1) We obtain a tighter bound on the cumulative gradient that depends on $t_0$, yielding an improved result for SGD without requiring strict assumptions under decaying learning rates, consistent with empirical results. (2) Our focus is on the effect of selection on generalization, so this task is only auxiliary and restricted to uniform averaging, while existing work considers weighted averaging schemes. In Appendix E, we provide the proofs of Lemma 4.4 and 4.10. The proofs of Theorems 4.6 and 4.11 are provided in Appendix F.2 and F.3.

## 5 EXPERIMENT

We systematically explore the effectiveness of our method across three distinct settings: behavior cloning, image classification, and text classification. Details of the experimental setup, including network architectures, hyperparameters, and additional results, are provided in Appendix B.

Table 2: Performance comparison of various methods on D4RL Gym tasks with $K = 20$. Each result is evaluated as the mean of 60 random rollouts, based on 3 independently trained models with 20 trajectories per model. Detailed results are presented in Table 3.

| | Task | Dataset | SGD | SWA | EMA | LAWA | Random | SeWA (Ours) |
|---|---|---|---|---|---|---|---|---|
| | Hopper | medium | 1245.039 | 1281.910 | 1302.400 | 1310.875 | 1312.166 | **1361.202** |
| | Hopper | medium-expert | 1460.785 | 1427.47 | 1373.268 | 1563.307 | 1482.012 | **1571.127** |
| | Walker2d | medium | 3290.248 | 3308.464 | **3420.257** | 3325.873 | 3324.557 | 3364.886 |
| K=20 | Walker2d | medium-expert | 3458.693 | 3588.176 | 3667.809 | 3557.925 | 3650.846 | **3673.804** |
| | Halfcheetah | medium | 4850.490 | 4913.549 | 4848.006 | 4974.041 | 4924.613 | **5071.051** |
| | Halfcheetah | medium-expert | 5015.689 | 5024.723 | 4957.194 | 4993.524 | 4988.816 | **5085.628** |
| | Average | | 3220.157 | 3257.382 | 3261.489 | 3287.591 | 3280.502 | **3354.616** |

## 5.1 BEHAVIOR CLONING

**Experimental Setups.** We conduct comprehensive evaluations using the widely adopted D4RL benchmark (Fu et al., 2020; Hu et al., 2024a), focusing on Gym-MuJoCo locomotion tasks. These tasks serve as standard benchmarks due to their well-defined structure, prevalence of near-optimal trajectories, and smooth reward functions, making them particularly suitable for assessing reinforcement learning algorithms. For evaluation, we employ cumulative reward as the primary metric.

**Baselines.** To evaluate SeWA, we compare it with established baselines: SGD-based pre-training, SWA (Izmailov et al., 2018), and EMA (Szegedy et al., 2016), all adapted for behavior cloning. EMA follows Kaddour (2022), using a 0.9 decay and updating every $K$ steps. SWA begins after 75% of training with a cosine annealing scheduler, averaging parameters every $K$ steps. We also include LAWA (Sanyal et al., 2023) and a Random baseline, both of which average $K$ checkpoints from the last $k = 1000$ pre-training steps. LAWA samples at intervals, Random samples randomly. LAWA, Random, and our SeWA use only these checkpoints for evaluation, without retraining. SGD, SWA, and EMA report final results from their respective training processes, ensuring fair comparison.

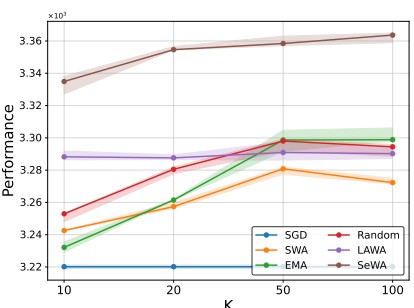

Figure 2: Comparison of different methods on the D4RL benchmark. Each data point represents the average cumulative reward across multiple tasks, averaged over 3 random seeds and 20 trajectories per seed. (Details in Appendix B.)

**Results.** In Figure 2 and Table 2, all baselines demonstrate superior performance compared to the original SGD optimizer, highlighting the effectiveness of weight averaging strategies in improving model performance. These results confirm that weight averaging can serve as a valuable technique for stabilizing and enhancing model training outcomes. Additionally, our analysis reveals that increasing the number of checkpoints $K$ used for averaging consistently improves performance across all methods. However, this improvement tends to plateau beyond a certain threshold, indicating diminishing returns as the number of averaged checkpoints increases. Notably, our SeWA consistently surpasses all baselines across experimental settings. Even with only $K = 10$ checkpoints, it outperforms baselines using $K = 100$, demonstrating both efficiency and robustness. This highlights our approach's efficiency and robustness, as it can deliver significant improvements with a substantially smaller computational footprint.

## 5.2 IMAGE CLASSIFICATION

**Experimental Setups.** We assess SeWA on image classification using the CIFAR-100 dataset and ResNet architecture (He et al., 2016). With 100 diverse classes, CIFAR-100 presents a challenging benchmark, and accuracy on the test set serves as our primary metric. In our experiments, we use intermediate model checkpoints saved during the final stage of training, specifically after 10,000 training steps. Performance is evaluated at intervals of $k = 100$ checkpoints, with the number of checkpoints included in the averaging procedure within each interval controlled by the hyperparameter $K$. This flexibility allows us to adjust the extent of checkpoint aggregation and analyze its impact.

**Results.** As illustrated in Figure 3, all baselines outperform the original SGD optimizer, underscoring the effectiveness of weight averaging in enhancing model performance. Additionally, weight averaging accelerates model convergence, with all baselines reaching performance levels that SGD requires

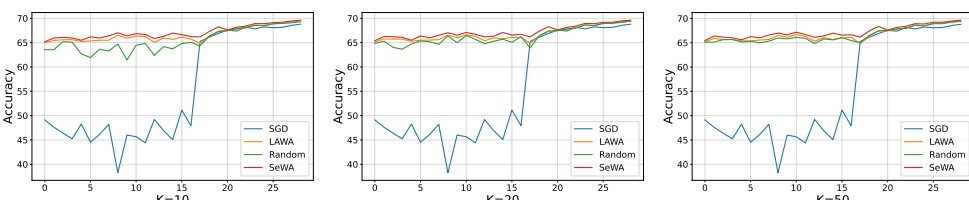

Figure 3: From left to right, the figures illustrate the impact of the hyperparameter $K$ on the CIFAR-100 task. Each point corresponds to intervals of 100 checkpoints, with $K$ checkpoints selected and averaged from these intervals using different strategies.

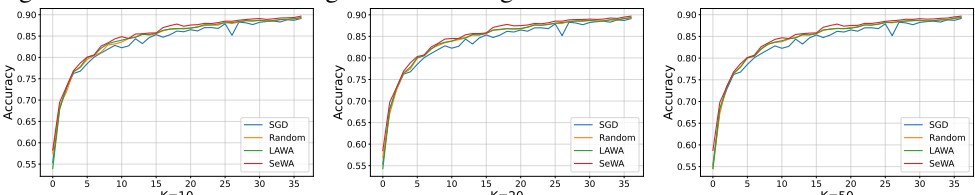

Figure 4: From left to right, the figures illustrate the impact of the hyperparameter $K$ on the AG News corpus. Each point corresponds to intervals of 100 checkpoints, with $K$ checkpoints selected and averaged from these intervals using different strategies.

17 steps to achieve. Our SeWA method consistently delivers the best performance, demonstrating its effectiveness. Beyond 17 steps, where the model approaches convergence, further improvement becomes minimal, as the checkpoints at this stage share highly similar weights.

### 5.3 TEXT CLASSIFICATION

**Experimental Setups.** For the text classification task, we use the AG News corpus, a widely used benchmark dataset containing news articles categorized into four distinct classes. The classification is performed using a transformer-based architecture Vaswani et al. (2017), which is known for its effectiveness in handling natural language processing tasks. To preprocess the dataset, we tokenize the entire corpus using the *basic_english* tokenizer. Any words not found in the vocabulary are replaced with a special token, *UNK*, to handle out-of-vocabulary terms. This preprocessing ensures that the dataset is standardized and ready for training. We save intermediate checkpoints throughout the training process, starting from the initial stages. From this set of checkpoints, we systematically select every $k = 100$ checkpoint for consideration in the averaging process. The hyperparameter $K$ controls the number of checkpoints used for averaging, allowing flexible experimentation with different levels of checkpoint aggregation. This experimental design facilitates a comprehensive evaluation of the effects of checkpoint averaging on model performance in NLP tasks.

**Results.** In Figure 4, the improvement of weight averaging over the SGD baseline is minimal for relatively simple tasks, primarily serving to stabilize training. However, our SeWA achieves the best results regardless of task complexity, demonstrating its broad applicability across diverse settings.

## 6 CONCLUSION

We propose a new algorithm SeWA for adaptive selecting checkpoints to average, which improves generalization and applies to a variety of tasks. In practical implementation, we employ probabilistic reparameterization to transform the discrete optimization problem into a continuous objective solvable by gradient-based methods. From a theoretical perspective, we prove that SeWA converges to a critical point with *flatter* curvature, thereby explaining its inherent ability to achieve better generalization. Moreover, under various assumptions, we derive its generalization bounds, which exhibit superior results compared to other algorithms. Empirically, we verify that SeWA can achieve good performance for unstable training processes, and a few checkpoints selected by SeWA can achieve results, while other algorithms require several times as many points.

**Limitation:** The theoretical analysis of SeWA based on $L$-Lipschitz and $\beta$-smoothness, which do not always hold in real-world deep learning models. Extending our framework through similar assumption-free analyses presents an interesting direction for future research.

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

**The Use of Large Language Models.** In this work, we exclusively employ large language models (LLMs) to refine the writing and presentation of our manuscript.

# A    ADDITIONAL RELATED WORK

**Weight averaging algorithm.** Model averaging methods, initially introduced in convex optimization Ruppert (1988); Polyak & Juditsky (1992); Li et al. (2023), have been widely used in various areas of deep learning and have shown their advantages in generalization and convergence. Subsequently, the introduction of SWA Izmailov et al. (2018), which averages the weights along the trajectory of SGD, significantly improves the model's generalization. Further modifications have been proposed, including the Stochastic Weight Average Density (SWAD) Cha et al. (2021), which averages checkpoints more densely, leading to the discovery of flatter minima associated with better generalization. Trainable Weight Averaging (TWA) Li et al. (2022) has improved the efficiency of SWA by employing trainable averaging coefficients. What's more, other approaches like Exponential Moving Average (EMA) Szegedy et al. (2016) and finite averaging algorithms, such as LAWA Kaddour (2022); Sanyal et al. (2023), which average the last $k$ checkpoints from running a moving window at a predetermined interval, employ different strategies to average checkpoints. These techniques have empirically shown faster convergence and better generalization. In meta-learning, Bayesian Model Averaging (BMA) is used to reduce the uncertainty of the model Huang et al. (2020). However, these algorithms often require manual design of averaging strategies and are only applicable to some specific tasks, imposing an additional cost on the training.

**Stability Analysis.** Stability analysis is a fundamental theoretical tool for studying the generalization ability of algorithms by examining their stability (Devroye & Wagner, 1979; Bousquet & Elisseeff, 2002; Mukherjee et al., 2006; Shalev-Shwartz et al., 2010). Based on this, Hardt et al. (2016) use the algorithm stability to derive generalization bounds for SGD, inspiring a series of works Charles & Papailiopoulos (2018); Zhou et al. (2018); Yuan et al. (2019); Lei & Ying (2020). This analysis framework has been extended to various domains, such as online learning (Yang et al., 2021), adversarial training (Xiao et al., 2022), decentralized learning (Zhu et al., 2023), and federated learning (Sun et al., 2023b;a). Although uniform sampling is a standard operation for building stability boundaries, selecting the initial point and sampling without replacement also significantly affects generalization and has been investigated in Shamir (2016); Kuzborskij & Lampert (2018). For the averaging algorithm, Hardt et al. (2016) and Xiao et al. (2022) analyze the generalization performance of SWA and establish stability bounds for the algorithm under the setting of convex and sampling with replacement. The primary focus of this paper is the construction of stability bounds for SeWA in both convex and non-convex settings.

**Mask Learning.** The general approach involves transforming the discrete optimization problem into a continuous one using probabilistic reparameterization, thereby enabling gradient-based optimization. Zhou et al. (2022) solves the coreset selection problem based on this by using a Policy Gradient Estimator (PGE) for a bilevel optimization objective. Zhang et al. (2024) propose a probabilistic masking method that improves diffusion model efficiency by skipping redundant steps. While the PGE method may suffer from high variance and unstable training, we solve the mask learning problem using the Gumbel-softmax reparameterization (Jang et al., 2017; Maddison et al., 2017). Mask learning has also been successfully applied across various domains to tackle diverse challenges (Liu et al., 2018; Hu et al., 2024c;b). In this paper, we aim to adaptively select checkpoints for model averaging, with the goal of improving generalization performance and mitigating training instability.

# B    EXPERIMENT DETAILS

## B.1    BEHAVIOR CLONING

**Network Architecture.** The network architecture comprises four layers, each consisting of a sequence of ReLU activation, Dropout for regularization, and a Linear transformation. The final layer includes an additional Tanh activation function to enhance the representation and capture non-linearities in the output.

**Results.** Comprehensive results for each task across all datasets are presented in Table 3. Our evaluation focuses specifically on the medium and medium-expert datasets, which offer a balanced

Table 3: Performance comparison of various methods on D4RL Gym tasks. The left panel shows results obtained using the final checkpoint under different update strategies, while the right panel presents results from averaged checkpoints collected during the final training stage with SGD, using different selection strategies. Each result is evaluated as the mean of 60 random rollouts, based on 3 independently trained models with 20 trajectories per model.

| | Task | Dataset | SGD | SWA | EMA | LAWA | Random | SeWA (Ours) |
|---|---|---|---|---|---|---|---|---|
| K=10 | Hopper | medium | 1245.039 | 1279.249 | 1297.270 | 1289.515 | 1291.478 | **1324.848** |
| | Hopper | medium-expert | 1460.785 | 1468.893 | 1320.408 | 1462.452 | 1451.015 | **1509.317** |
| | Walker2d | medium | 3290.248 | 3328.121 | 3341.888 | 3341.437 | 3306.763 | **3371.202** |
| | Walker2d | medium-expert | 3458.693 | 3546.008 | 3681.504 | 3634.373 | 3609.611 | **3679.806** |
| | Halfcheetah | medium | 4850.490 | 4858.224 | 4894.204 | 5012.389 | 4896.104 | **5041.369** |
| | Halfcheetah | medium-expert | 5015.689 | 4974.923 | 4857.562 | 4989.329 | 4962.719 | **5082.902** |
| | Average | | 3220.157 | 3242.570 | 3232.139 | 3288.249 | 3252.948 | **3334.907** |
| K=20 | Hopper | medium | 1245.039 | 1281.910 | 1302.400 | 1310.875 | 1312.166 | **1361.202** |
| | Hopper | medium-expert | 1460.785 | 1427.47 | 1373.268 | 1563.307 | 1482.012 | **1571.127** |
| | Walker2d | medium | 3290.248 | 3308.464 | **3420.257** | 3325.873 | 3324.557 | 3364.886 |
| | Walker2d | medium-expert | 3458.693 | 3588.176 | 3667.809 | 3557.925 | 3650.846 | **3673.804** |
| | Halfcheetah | medium | 4850.490 | 4913.549 | 4848.006 | 4974.041 | 4924.613 | **5071.051** |
| | Halfcheetah | medium-expert | 5015.689 | 5024.723 | 4957.194 | 4993.524 | 4988.816 | **5085.628** |
| | Average | | 3220.157 | 3257.382 | 3261.489 | 3287.591 | 3280.502 | **3354.616** |
| K=50 | Hopper | medium | 1245.039 | 1294.884 | 1329.863 | 1336.33 | 1319.571 | **1389.280** |
| | Hopper | medium-expert | 1460.785 | 1477.466 | 1485.696 | 1537.672 | 1496.045 | **1616.116** |
| | Walker2d | medium | 3290.248 | 3262.046 | 3341.767 | 3253.695 | 3352.12 | **3392.130** |
| | Walker2d | medium-expert | 3458.693 | 3577.509 | 3591.081 | 3584.468 | 3659.789 | **3672.560** |
| | Halfcheetah | medium | 4850.490 | 4927.951 | 4968.048 | 5022.097 | 5000.004 | **5035.631** |
| | Halfcheetah | medium-expert | 5015.689 | 5061.688 | **5075.426** | 5011.232 | 4960.585 | 5044.886 |
| | Average | | 3220.157 | 3280.833 | 3298.647 | 3290.916 | 3298.019 | **3358.434** |
| K=100 | Hopper | medium | 1245.039 | 1347.267 | 1322.625 | 1320.652 | 1319.727 | **1393.981** |
| | Hopper | medium-expert | 1460.785 | 1527.206 | 1528.265 | 1496.266 | 1491.196 | **1568.025** |
| | Walker2d | medium | 3290.248 | 3324.218 | 3393.646 | 3345.913 | 3321.046 | **3424.078** |
| | Walker2d | medium-expert | 3458.693 | 3575.621 | 3629.308 | 3613.274 | 3587.211 | **3710.347** |
| | Halfcheetah | medium | 4850.490 | 4939.629 | 4871.376 | 4974.220 | 5015.349 | **5021.948** |
| | Halfcheetah | medium-expert | 5015.689 | 4919.624 | 5047.757 | 4991.007 | 5031.975 | **5063.546** |
| | Average | | 3220.157 | 3272.261 | 3298.830 | 3290.222 | 3294.417 | **3363.654** |

mix of trajectories with varying performance levels. This selection enables a thorough assessment of our method's ability to generalize across different reward distributions. For clarity and ease of comparison, the main paper emphasizes the average performance across tasks, as illustrated in Figure 2. This dual presentation ensures a detailed examination of individual tasks while providing an accessible overview of overall performance.

## B.2 IMAGE CLASSIFICATION OF CIFAR100

**Network Architecture.** The network architecture consists of three primary blocks, followed by an average pooling layer and a linear layer for generating the final output. Each block contains two convolutional layers, each accompanied by a corresponding batch normalization layer to improve training stability and convergence. To address potential issues of vanishing gradients, each block includes a shortcut connection that facilitates efficient gradient flow during backpropagation. The output of each block is passed through a ReLU activation function to introduce non-linearity, enabling the network to learn complex representations effectively.

**Results.** In addition to the results presented in Figure 3, we provide further analysis examining the impact of network parameter variations to demonstrate the robustness of our method across networks of different sizes. These results, shown in Figure 5, illustrate that as the number of layers or blocks increases, the performance of SGD improves, following a similar training curve.

Notably, weight averaging consistently outperforms SGD during the upward phase of training. The performance gains from weight averaging become more pronounced as the network size increases, highlighting its potential in scaling effectively to larger models. This highlights the potential of weight averaging to enhance the performance of larger models. Furthermore, regardless of changes

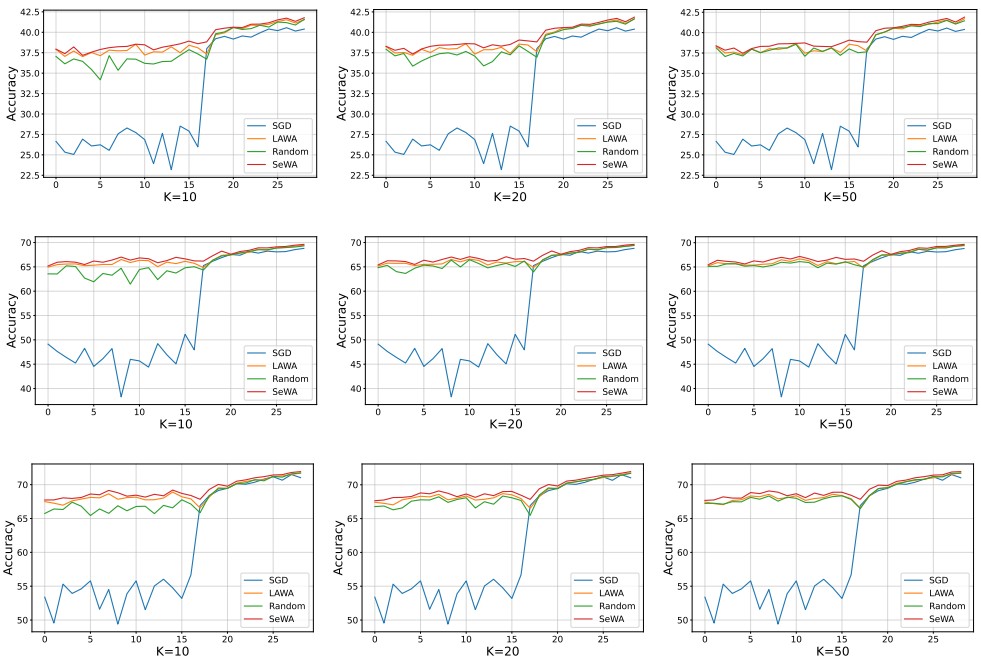

Figure 5: From left to right, the figures illustrate the impact of the hyperparameter $K$ on the CIFAR-100 task. Each data point represents performance based on intervals of $k = 100$ checkpoints, with $K$ checkpoints selected from these intervals using various strategies. The first row corresponds to a network architecture with 1 block, the second row represents a network with 3 blocks, and the third row depicts results for a network with 5 blocks.

in network parameters, our proposed method consistently achieves superior results, demonstrating its adaptability and effectiveness across varying network configurations. These findings emphasize the potential of weight averaging as a robust and scalable technique for optimizing model performance.

### B.2.1 COMPARISON OF DIFFERENT METHODS ON THE CIFAR100

We conduct experiments on CIFAR100 using the VGG16 network, and compare with SGD, SWA, LAWA, and EMA. The SGD achieves an accuracy of 72.9%. In the following table, we present the accuracy improvements of each method over SGD. It shows that our method achieves considerable gain over other averaging methods.

Table 4: Comparison of different methods on the Cifar100.

|  | $K = 10$ | $K = 20$ | $K = 30$ | $K = 40$ | $K = 50$ |
|---|---|---|---|---|---|
| SWA | -0.05 | +0.03 | +0.09 | +0.12 | +0.06 |
| LAWA | -0.01 | +0.01 | +0.01 | +0.05 | +0.04 |
| EMA | -0.10 | +0.08 | +0.06 | 0.00 | -0.20 |
| SeWA | +0.26 | +0.26 | +0.41 | +0.51 | +0.49 |

As shown in the above table, our method shows considerable gain over other averaging methods, including SWA and EMA. In the image classification task, the training dynamics are very stable; therefore, we think that averaging methods are not so significant here. For example, SWA, LAWA, and EMA do not improve the performance much. Therefore, when the base model performs well and training proceeds steadily, limited improvement is not a weakness unique to the SeWA algorithm in such tasks, but rather a characteristic shared by all algorithms.

### B.3 Image Classification of ImageNet

**Experimental Setups.** To rigorously evaluate our method's efficacy in image classification, we employ the ImageNet dataset Deng et al. (2009) in conjunction with the Vision Transformer (ViT) architecture Han et al. (2022). The ImageNet dataset, comprising 1000 diverse classes, serves as a comprehensive benchmark for assessing image classification performance. We adopt classification accuracy on the test dataset as our primary evaluation metric. Throughout our experimental protocols, we systematically preserve model checkpoints after each training epoch. Performance evaluation is conducted at intervals of $k = 5$ checkpoints, with the number of checkpoints incorporated into the averaging procedure within each interval regulated by the hyperparameter $K = 3$.

**Network Architecture.** Our implementation utilizes a ViT model (330.23MB), representing a paradigm shift from conventional convolutional neural networks for image classification tasks. The ViT architecture initially employs a patch embedding layer that segments input images into uniform patches and projects them into a high-dimensional embedding space. A learnable classification token is subsequently prepended to the sequence of embedded patches, and positional embeddings are incorporated to preserve spatial information. The architectural core comprises 12 transformer blocks, each integrating multi-head self-attention mechanisms with 12 attention heads and feed-forward networks with an expansion ratio of 4. The resultant representations undergo normalization via layer normalization before transmission to a linear classification head that generates output logits corresponding to the 1000 ImageNet classes.

**Results.** We present a comprehensive analysis examining the efficacy of various weight averaging strategies when applied to transformer-based architectures. The empirical results, illustrated in Figure 6, demonstrate that our proposed SeWA consistently outperforms standard SGD optimization throughout the training trajectory.

Significantly, all weight averaging methods demonstrate superior accuracy compared to SGD throughout training. These findings highlight the particular effectiveness of weight averaging. Moreover, while Random weight averaging generally outperforms SGD, it shows inferior results compared to our proposed SeWA and occasionally underperforms relative to SGD. In contrast, our approaches maintain consistent performance advantages throughout the learning process. This comparative analysis provides compelling evidence that structured weight averaging substantially enhances Vision Transformer performance on large-scale image classification tasks. The demonstrated superiority of our methodologies over both baseline SGD and Random underscores the importance of the adaptive selection process in optimizing transformer networks, showing the effectiveness of our method.

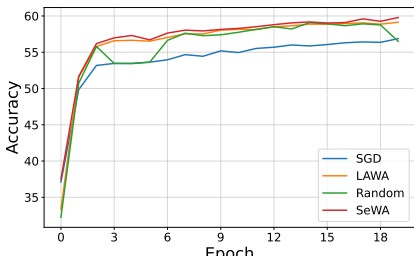

Figure 6: Comparison of different methods on the ImageNet benchmark utilizing the ViT architecture.

### B.4 Text Classification

**Network Architectures.** The network architecture comprises two embedding layers followed by two layers of *TransformerEncoderLayer*. Each *TransformerEncoderLayer* includes a multi-head self-attention mechanism and a position-wise feedforward network, along with layer normalization and residual connections to enhance training stability and gradient flow. The output from the Transformer layers is passed through a linear layer to produce the final predictions.

**Results.** In addition to the findings presented in Figure 4, we conduct further analysis to evaluate the impact of network parameter variations, demonstrating the robustness of our method across networks of varying sizes. These additional results, shown in Figure 7, indicate that as the number of Transformer layers increases, the performance of SGD improves up to a certain point. However, beyond this range - where two layers appear sufficient - performance begins to exhibit fluctuations, suggesting diminishing returns and instability with additional layers.

While the improvement achieved by weight averaging is relatively modest due to the simplicity of the task, it still plays a critical role in stabilizing the training process and reducing fluctuations in the training curve. Among the averaging methods evaluated, our proposed method consistently

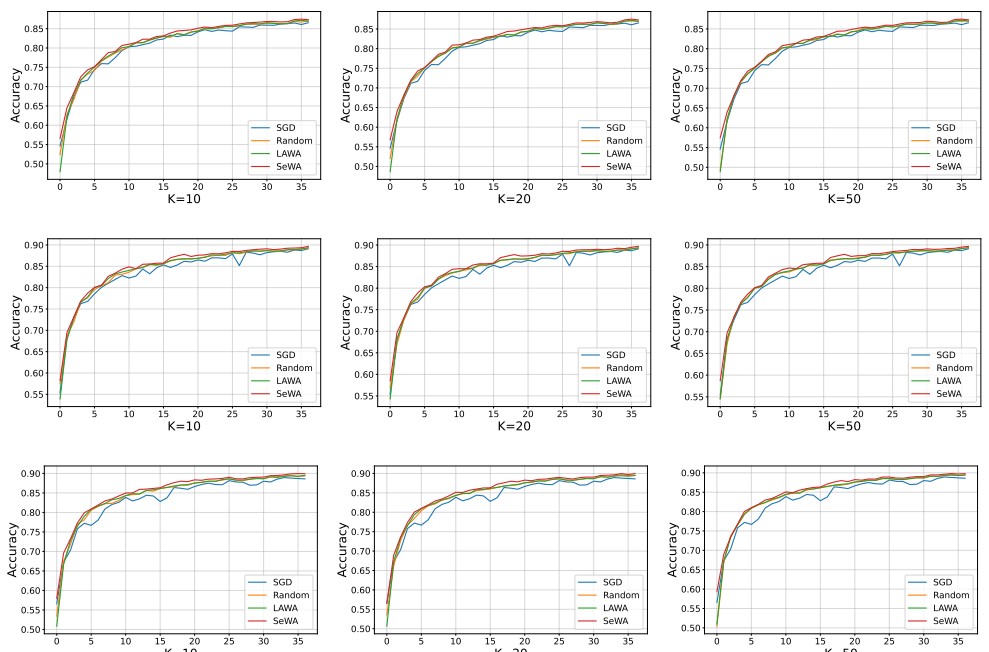

Figure 7: From left to right, the figures illustrate the impact of the hyperparameter $K$ on the AG News corpus. Each point corresponds to intervals of $k = 100$ checkpoints, with $K$ checkpoints selected from these intervals using different strategies. The first row corresponds to a network architecture with a single *TransformerEncoderLayer*, the second row represents a network with three *TransformerEncoderLayer*s, and the third row shows results for a network with five *TransformerEncoderLayer*s.

achieves the best performance, underscoring its effectiveness in maintaining stability and optimizing performance, even in scenarios where task complexity is low.

## B.5 EXTENDED EXPERIMENTS ON THE NYUV2 DATASET

To further verify whether the SeWA algorithm fails in specific tasks, we additionally compared the SGD and SeWA algorithms in a regression task. We conducted extended experiments on the NYUv2 dataset using the ViT model, achieving average improvements of +4.89% on the training set and +1.88% on the validation set.

Table 5: Comparison of the SGD and SeWA on the NYUv2 dataset.

| Method | 2300 | 4300 | 6300 | 8300 (iterations) |
|---|---|---|---|---|
| SGD (Train) | 0.1668 | 0.1343 | 0.1237 | 0.1235 |
| SeWA (Train) | 0.1463 | 0.1324 | 0.1207 | 0.1160 |
| SGD (Val) | 0.1970 | 0.1719 | 0.1683 | 0.1680 |
| SeWA (Val) | 0.1813 | 0.1689 | 0.1674 | 0.1648 |

In extended experiments, we reached identical conclusions. First, as a post-training algorithm, SeWA enhances the generalization capability of the base model even when the base model has already converged. Second, due to the relatively stable task training process, the SeWA algorithm demonstrates equally marginal improvements when applied to additional regression problems. However, in behavior cloning experiments (Section 5.1), where the training process is significantly more unstable, our method demonstrates even more pronounced improvements.

## B.6 ABLATIONS

### B.6.1 HYPERPARAMETER SENSITIVITY

Our proposed algorithm introduces several new hyperparameters, and understanding their impact is critical for both reproducibility and practical deployment. To this end, we perform comprehensive sensitivity analyses on representative tasks, focusing primarily on the `Hopper-medium` and `Hopper-medium-expert` environments from D4RL. All reported results are averaged over three random seeds with 20 evaluations per seed, and we present both mean and standard deviation.

Table 6: Ablation on the Gumbel-Softmax temperature $t$ in Algorithm 2.1.

| $t$ | 0.1 | 0.3 | 0.5 | 0.7 | 1.0 |
|---|---|---|---|---|---|
| Hopper-medium | $1372.11 \pm 48.01$ | $1377.29 \pm 47.12$ | $1384.03 \pm 46.56$ | $1384.45 \pm 47.02$ | $1389.28 \pm 46.98$ |
| Hopper-medium-expert | $1595.42 \pm 12.51$ | $1601.23 \pm 11.87$ | $1609.01 \pm 11.35$ | $1610.65 \pm 10.72$ | $1616.12 \pm 10.58$ |

**Effect of Gumbel-Softmax Temperature $t$.** Table 6 reports the performance of SeWA under different Gumbel-Softmax temperature values $t$. The results demonstrate that SeWA is generally robust over a wide range of temperatures. Performance degradation is observed only at very small temperatures (e.g., $t = 0.1$), where the Gumbel-Softmax distribution becomes nearly discrete, resulting in high-variance gradients and challenging optimization. For moderate and large $t$, performance remains stable and exhibits small variance, indicating that SeWA does not require fine-grained tuning of this hyperparameter.

Table 7: Ablation on the number of MC samples $M$ in Algorithm 2.1

| $M$ | 1 | 5 | 10 | 20 |
|---|---|---|---|---|
| wall-clock time | 0.25 s/iter | 0.31 s/iter | 0.33 s/iter | 0.36 s/iter |
| Hopper-medium | $1389.28 \pm 46.98$ | $1390.52 \pm 37.06$ | $1399.12 \pm 30.95$ | $1414.08 \pm 8.86$ |
| Hopper-medium-expert | $1616.12 \pm 10.58$ | $1621.11 \pm 9.33$ | $1625.15 \pm 7.95$ | $1632.37 \pm 4.12$ |

**Effect of Monte Carlo Sample Size $M$.** We further examine the impact of the number of Monte Carlo (MC) samples $M$ used for gradient estimation. In this ablation, we fix $K = 50$ (number of selected checkpoints) and vary $M$ from 1 to 20. Table 7 shows that increasing $M$ consistently reduces performance variance and yields slightly improved returns, which is expected as a result of more accurate gradient estimation. Importantly, the computational overhead grows only mildly - using $20\times$ samples results in merely $1.44\times$ wall-clock time - making higher $M$ values computationally feasible. This suggests that practitioners can choose $M$ flexibly based on their computational budget: larger $M$ improves performance but is not strictly necessary to achieve strong results.

Table 8: Ablation on the max iterations for mask optimization in Algorithm 2.1

| max iteration | 100 | 500 | 1000 | 1500 |
|---|---|---|---|---|
| Hopper-medium | $1370.46 \pm 48.90$ | $1381.39 \pm 47.23$ | $1389.28 \pm 46.98$ | $1390.01 \pm 46.92$ |
| Hopper-medium-expert | $1590.51 \pm 12.33$ | $1607.22 \pm 11.04$ | $1616.12 \pm 10.58$ | $1616.53 \pm 10.56$ |

**Effect of Maximum Iterations for Mask Optimization.** We next investigate the influence of the maximum number of iterations used in the mask optimization step. As shown in Table 8, performance improves as the number of iterations increases, but the gain saturates at approximately 1000 iterations. This indicates that SeWA converges quickly and does not require excessively long optimization schedules to achieve near-optimal performance - an important property for practical efficiency.

**Effect of Candidate Pool Size $k$.** Finally, we examine the candidate pool size $k$, i.e., the number of recent checkpoints retained in $\mathbf{w}$. Table 9 shows that increasing $k$ improves performance marginally, with diminishing returns beyond $k = 1000$. Notably, even small pool sizes (e.g., $k = 500$) lead to competitive results, suggesting that SeWA can be deployed efficiently without excessive memory requirements for checkpoint storage.

Table 9: Ablation on the candidate pool size $k$ (len($\mathbf{w}$)) in Algorithm 2.1

| pool $k$ | 100 | 500 | 1000 | 1500 |
|---|---|---|---|---|
| Hopper-medium | $1362.88 \pm 49.45$ | $1375.20 \pm 47.89$ | $1389.28 \pm 46.98$ | $1389.30 \pm 47.10$ |
| Hopper-medium-expert | $1581.92 \pm 13.42$ | $1603.11 \pm 11.58$ | $1616.12 \pm 10.58$ | $1616.80 \pm 10.71$ |

### B.6.2 EXTENDED ANALYSIS OF AVERAGING STRATEGIES

Table 10: Ablation study on averaging strategies for checkpoint selection.

| | SeWA | All-k-Average | Top-K-Average |
|---|---|---|---|
| Hopper-medium | $1389.28 \pm 46.98$ | $1285.38 \pm 42.33$ | $1356.18 \pm 40.44$ |
| Hopper-medium-expert | $1616.12 \pm 10.58$ | $1483.20 \pm 10.33$ | $1533.28 \pm 9.32$ |

To provide a comprehensive evaluation of our averaging methodology, we conduct additional ablation studies examining alternative averaging strategies. Specifically, we compare SeWA against two baseline approaches: All-k-Average, which computes the arithmetic mean of all candidate checkpoints (where $k = 1000$ for both *Hopper-medium* and *Hopper-medium-expert* environments), and Top-K-Average ($K = 50$), which requires evaluating all $k$ candidate points before selecting and averaging the top-performing subset. Due to the computational overhead associated with evaluating all $k$ checkpoints in the Top-K-Average approach, we limit this analysis to two representative environments.

The experimental results presented in Table 10 reveal several important insights. The All-k-Average strategy demonstrates inferior performance compared to SeWA, which can be attributed to information dilution effects. By indiscriminately averaging all candidate checkpoints, this approach fails to prioritize high-quality solutions and incorporates potentially detrimental weights from suboptimal checkpoints, ultimately leading to degraded performance.

Similarly, the Top-K-Average method yields lower performance than SeWA, despite its computational expense in evaluating all candidate points. These findings provide compelling evidence that SeWA's effectiveness stems from its ability to identify and leverage checkpoints that genuinely contribute to improved target performance, rather than simply aggregating high-performing individual models. The results demonstrate that not all high-performance checkpoints are conducive to exploring flat regions of the loss landscape when combined through averaging. This observation underscores the critical importance of SeWA's intelligent selection mechanism in the averaging process, which goes beyond naive performance-based selection to identify checkpoints that exhibit beneficial geometric properties when aggregated.

### B.6.3 COMPARISON UNDER EQUAL CALCULATIONS

Unlike prior approaches that rely on extensive hyperparameter sweeps (e.g., determining which models to average), SeWA introduces a lightweight post-hoc mask optimization procedure (Algorithm 2.1). This additional phase incurs only modest computational overhead-approximately 10% of the pretraining cost (e.g., 1,000 vs. 10,000 steps)-while substantially reducing manual tuning effort.

To further address the equal-compute comparison, we performed an SGD-extended baseline in which SGD is allowed to continue training for the same number of extra steps as SeWA, including all evaluations of $F$ during mask optimization. Even under this enlarged budget, the SGD-extended baseline consistently underperforms SeWA, as shown below.

Table 11: Ablation study on equal calculations.

| | SGD | $\text{SGD}_{enhanced}$ | SeWA |
|---|---|---|---|
| Hopper-medium | $1245.04 \pm 69.32$ | $1275.34 \pm 67.31$ | $1389.28 \pm 46.98$ |
| Hopper-medium-expert | $1460.79 \pm 15.31$ | $1480.85 \pm 15.22$ | $1616.12 \pm 10.58$ |

Although selecting the best single SGD checkpoint provides a competitive baseline, it typically exhibits high variance and limited robustness across tasks. Extending training alone does not reliably

improve performance, as SGD often oscillates near local minima in later stages. In contrast, SeWA adaptively aggregates informative checkpoints, stabilizing the optimization trajectory and achieving consistently better generalization with far fewer averaged points. These results indicate that SeWA's improvements stem not from additional computation but from its principled checkpoint selection strategy, which enhances both stability and generalization.

### B.7 COMPARISON WITH TWA ALGORITHM

Now, we are also training using the TWA approach in the tail phase of the training process, which involves learning a set of continuous weight coefficients. We conduct experiments of training a VGG16 network on the CIFAR-10 dataset, following the setting in the TWA paper. We apply SeWA on the 100 tail training checkpoints, learning masks for an additional 10 epochs, which is the same with TWA. The number of chosen checkpoints is set to $K \in \{10, 20, 30\}$, and the MC samples is set to $M = 1$, which makes the training cost identical to TWA. The results are shown in the following table. Our method tends to outperform TWA.

Table 12: Comparison with TWA algorithm.

| TWA | SeWA (K=10) | SeWA (K=20) | SeWA (K=30) |
|---|---|---|---|
| $94.16 \pm 0.14$ | $94.26 \pm 0.02$ | $94.30 \pm 0.02$ | $94.28 \pm 0.05$ |

With the current settings, both SeWA and TWA improve model generalization by averaging collected checkpoints. While TWA employs a finer-grained approach to utilize information from all parameters, SeWA aims to exclude redundant parameters through masking, retaining only representative ones. Furthermore, SeWA's computational overhead primarily stems from the masking learning process, which occurs only after the base training is completed. In contrast, TWA performs additional orthogonalization during each parameter update, incurring this extra computational cost throughout the entire training process.

## C PROOF OF LEMMA 2.4

$(1 + \alpha\beta)$-**expansive.** According to triangle inequality and $\beta$-smoothness,

$$\begin{aligned} \|w_{T+1} - w'_{T+1}\| &\leq \|w_T - w'_T\| + \alpha\|\nabla F(w_T) - \nabla F(w'_T)\| \\ &\leq \|w_T - w'_T\| + \alpha\beta\|w_T - w'_T\| \\ &= (1 + \alpha\beta)\|w_T - w'_T\|. \end{aligned} \tag{15}$$

***Non*-expansive.** Function is convexity and $\beta$-smoothness that implies

$$\langle \nabla F(w) - \nabla F(v), w - v \rangle \geq \frac{1}{\beta}\|\nabla F(w) - \nabla F(v)\|^2. \tag{16}$$

We conclude that

$$\begin{aligned} \|w_{T+1} - w'_{T+1}\| &= \sqrt{\|w_T - \alpha\nabla F(w_T) - w'_T + \alpha\nabla F(w'_T)\|^2} \\ &= \sqrt{\|w_T - w'_T\|^2 - 2\alpha\langle\nabla F(w_T) - \nabla F(w'_T), w_T - w'_T\rangle + \alpha^2\|\nabla F(w_T) - \nabla F(w'_T)\|^2} \\ &\leq \sqrt{\|w_T - w'_T\|^2 - \left(\frac{2\alpha}{\beta} - \alpha^2\right)\|\nabla F(w_T) - \nabla F(w'_T)\|^2} \\ &\leq \|w_T - w'_T\|. \end{aligned} \tag{17}$$

## D PROOF OF THEOREM 4.2

We consider the new auxiliary function $G(s) \triangleq -F(s)$. By the smoothness of the function $F$, we know that the auxiliary function $G$ is also $\beta$-smoothness such that

$$G(s_{t+1}) \geq G(s_t) + \langle\nabla G(s_t), s_{t+1} - s_t\rangle - \frac{\beta}{2}\|s_{t+1} - s_t\|^2.$$

Note that, in gradient descent, we have $s_{t+1} = \mathcal{P}_C\left(s_t + \mu_t \nabla G(s_t)\right)$ where $\mu_t$ is learning rate and thus using the properties of convex projections we have

$$\langle s_{t+1} - s_t, s_{t+1} - (s_t + \mu_t \nabla G(s_t))\rangle \leq 0 \quad \Rightarrow \quad \|s_t - s_{t+1}\|^2 \leq \mu_t \langle s_{t+1} - s_t, \nabla G(s_t)\rangle.$$

Plugging this into the latter inequality we conclude that for $\mu_t \leq \frac{1}{\beta}$

$$G(s_{t+1}) \geq G(s_t) + \left(\frac{1}{\mu_t} - \frac{\beta}{2}\right) \|s_{t+1} - s_t\|^2 \geq G(s_t) + \frac{\beta}{2}\|s_{t+1} - s_t\|^2.$$

Summing both sides we conclude that

$$\sum_{t=1}^{\infty} \|s_{t+1} - s_t\|^2,$$

is bounded, which implies that $s_t$ converges to a point $s$. This means that this point obeys

$$s = \mathcal{P}_C\left(s + \mu_t \nabla G(s)\right).$$

By definition of projection the latter implies that $\mathcal{P}_{C-\{s\}}(\mu_t \nabla G(s)) = 0$. A well known result in convex analysis (Rockafellar, 2015) implies $\max_{y \in C}\langle \nabla G(s), y - s\rangle = -\min_{y \in C}\langle \nabla F(s), y - s\rangle \leq 0$, concluding the proof.

# E   PROOF OF THE LEMMA 4.4 AND 4.10

We establish generalization bounds for the SeWA algorithm through uniform stability (Eq. 4). Let $\bar{w}_T^K$ and $\bar{w}_T^{K\prime}$ denote the SeWA's outputs under perturbations arising from two sources: (1) data perturbation, where SeWA runs on two datasets $S$ and $S'$ differing by exactly one sample; (2) weight selection for averaging, an inherent algorithmic procedure. To analyze this, we first fix the selected weights, ensuring identical selection probabilities at each step $i$. We can apply stability theory to achieve our research objectives based on the above.

## E.1   PROOF OF LEMMA 4.4

We now fix an example $z$ and use the Lipschitz assumption, which transforms the problem into bounding the parameter differences.

$$\mathbb{E}_{z,m,A}|F(\bar{w}_T^K; z) - F(\bar{w}_T^{K\prime}; z)| \leq L\mathbb{E}_{m,A}\|\bar{w}_T^K - \bar{w}_T^{K\prime}\|$$

$$\leq L\left(\frac{1}{k}\sum_{i=T-k+1}^{T} s_i \mathbb{E}_A\|w_i - w_i'\| + \frac{1}{k}\sum_{i=T-k+1}^{T}(1 - s_i)\cdot 0\right)$$

$$\leq \hat{s}L\mathbb{E}_A[\bar{\delta}_T],$$

where the second inequality is based on taking the expectation for mask $m_i$, and the last inequality is because of $\hat{s} = \sup_{T-k+1 \leq i \leq T} s_i$.

## E.2   PROOF OF LEMMA 4.10

There are differences in the proof between convex and nonconvex assumptions.

We split the proof of Lemma 4.10 into two parts. Let $\xi$ denote the event $\bar{\delta}_{t_0} = 0$. Let $z$ be an arbitrary example and consider the random variable $I$ assuming the index of the first time step using the different sample. Then we have

$$\mathbb{E}|\nabla F(\bar{w}_T^K; z) - \nabla F(\bar{w}_T^{K\prime}; z)| = P\{\xi\}\,\mathbb{E}[|\nabla F(\bar{w}_T^K; z) - \nabla F(\bar{w}_T^{K\prime}; z)||\xi]$$

$$+ P\{\xi^c\}\,E[|\nabla F(\bar{w}_T^K; z) - \nabla F(\bar{w}_T^{K\prime}; z)||\xi^c]$$

$$\leq P\{I \geq t_0\} \cdot \mathbb{E}[|\nabla F(\bar{w}_T^K; z) - \nabla F(\bar{w}_T^{K\prime}; z)||\xi] \qquad (18)$$

$$+ P\{I \leq t_0\} \cdot \sup_{\bar{w}^K, z} F(\bar{w}^K; z),$$

where $\xi^c$ denotes the complement of $\xi$.

Note that when $I \geq t_0$, then we must have that $\bar{\delta}_{t_0} = 0$, since the execution on $S$ and $S'$ is identical until step $t_0$. We can get $LE[\|\bar{w}_T^K - \bar{w}_T^{K'}\| | \xi]$ combined the Lipschitz continuity of $F$. Furthermore, we know $P\{\xi^c\} = P\{\bar{\delta}_{t_0} = 0\} \leq P\{I \leq t_0\}$, for the random selection rule, we have

$$P\{I \leq t_0\} \leq \sum_{t=1}^{t_0} P\{I = t_0\} = \frac{t_0}{n}. \tag{19}$$

We can combine the above two parts and $F \in [0, 1]$ to derive the stated bound

$$\mathbb{E}|F(\bar{w}_T^K; z) - F(\bar{w}_T^{K'}; z)| \leq \frac{t_0}{n} + L\mathbb{E}\left[\|\bar{w}_T^K - \bar{w}_T^{K'}\| \|\bar{w}_{t_0}^K - \bar{w}_{t_0}^{K'}\| = 0\right]. \tag{20}$$

Secondly, we take expectation for the $m_i$ of $\mathbb{E}\|\bar{w}_T^K - \bar{w}_T^{K'}\|$, which is similar to the proof of Lemma 4.4. Then we have

$$\mathbb{E}|F(\bar{w}_T^K; z) - F(\bar{w}_T^{K'}; z)| \leq \frac{t_0}{n} + \hat{s}L\mathbb{E}[\bar{\delta}_T | \bar{\delta}_{t_0} = 0], \tag{21}$$

where $\bar{\delta}_T = \frac{1}{k}\sum_{i=T-k+1}^{T} \|w_i - w_i'\|$, $w_i$ and $w_i'$ are the outputs of SGD, and $\hat{s} = \sup_{T-k+1 \leq i \leq T} s_i$, where $s_i$ is the probability of $m_i = 1$ and $\hat{s} \in (0, 1]$.

# F    PROOF OF THE GENERALIZATION BOUNDS

By the Lemma 4.4 and 4.10, the proof of Theorem 4.6 and 4.11 can be further decomposed into bounding the difference of the parameters for the last $k$ points of the average algorithm.

## F.1    UPDATE RULES OF THE LAST $k$ POINTS OF THE AVERAGING ALGORITHM.

For the last $k$ points of the averaging algorithm, we formulate it as

$$\hat{w}_T^k = \frac{1}{k}\sum_{i=T-k+1}^{T} w_i. \tag{22}$$

It is not difficult to find the relationship between $\hat{w}_T^k$ and $\hat{w}_{T-1}^k$, i.e.,

$$\hat{w}_T^k = \hat{w}_{T-1}^k + \frac{1}{k}(w_T - w_{T-k}) = \hat{w}_{T-1}^k - \frac{1}{k}\sum_{i=T-k+1}^{T} \alpha_i \nabla F(w_{i-1}, z_i), \tag{23}$$

where the second equality follows from the update of SGD.

## F.2    PROOF. THEOREM 4.6

We finish the task using Lemma 4.4 and Lemma 4.10, which divide the task of establishing the SeWA's generalization bound into two parts: (1) analyzing the impact of the selection process, and (2) deriving the bound for averaging over the last $k$ points. Then, we first establish the generalization bound for averaging over the last $k$ points.

First, using the relationship between $\hat{w}_T^k$ and $\hat{w}_{T-1}^k$ in Eq. 23, we consider that the different sample $z_T$ and $z_T'$ are selected to update with probability $\frac{1}{n}$ at the step $T$.

$$\bar{\delta}_T = \bar{\delta}_{T-1} + \frac{1}{k}\sum_{i=T-k+1}^{T} \alpha_i \|\nabla F(w_{i-1}', z_i) - \nabla F(w_{i-1}, z_i)\|$$

$$\leq \bar{\delta}_{T-1} + \frac{2\alpha_T L}{k} + \frac{1}{k}\sum_{i=T-k+1}^{T-1} \alpha_i \|\nabla F(w_{i-1}', z_i) - \nabla F(w_{i-1}, z_i)\|, \tag{24}$$

where the proof follows from the triangle inequality and the $L$-Lipschitz condition. For $\frac{1}{k}\sum_{i=T-k+1}^{T-1} \alpha_i \|\nabla F(w_{i-1}', z_i) - \nabla F(w_{i-1}, z_i)\|$ will be controlled later.

Second, another situation needs to be considered in case of the same sample are selected($z_T = z'_T$) to update with probability $1 - \frac{1}{n}$ at the step $T$.

$$
\begin{aligned}
\bar{\delta}_T &= \bar{\delta}_{T-1} + \frac{1}{k} \sum_{i=T-k+1}^{T} \alpha_i \|\nabla F(w'_{i-1}, z_i) - \nabla F(w_{i-1}, z_i)\| \\
&\leq \bar{\delta}_{T-1} + \frac{1}{k} \sum_{i=T-k+1}^{T-1} \alpha_i \|\nabla F(w'_{i-1}, z_i) - \nabla F(w_{i-1}, z_i)\|,
\end{aligned}
\tag{25}
$$

where the second inequality comes from the non-expansive property of convex function.

For each $\|\nabla F(w'_{i-1}, z_i) - \nabla F(w_{i-1}, z_i)\|$ in the sense of expectation, We consider two situations using $\alpha L$ bound and the non-expansive property. Then, we get

$$
\frac{1}{k} \sum_{i=T-k+1}^{T-1} \alpha_i \|\nabla F(w'_{i-1}, z_i) - \nabla F(w_{i-1}, z_i)\| \leq \frac{2L}{nk} \sum_{i=T-k+1}^{T-1} \alpha_i.
\tag{26}
$$

Then we obtain the expectation based on the above analysis

$$
\begin{aligned}
\mathbb{E}\left[\bar{\delta}_T\right] &\leq (1 - \frac{1}{n})\bar{\delta}_{T-1} + \frac{1}{n}\left(\bar{\delta}_{T-1} + \frac{2\alpha_T L}{k}\right) + \frac{2L}{nk} \sum_{i=T-k+1}^{T-1} \alpha_i \\
&\leq \mathbb{E}\left[\bar{\delta}_{T-1}\right] + \frac{2L}{nk} \sum_{i=T-k+1}^{T} \alpha_i
\end{aligned}
\tag{27}
$$

recursively, we can get

$$
\begin{aligned}
\mathbb{E}\left[\bar{\delta}_T\right] &\leq \frac{2L}{nk}\left(\sum_{i=T-k+1}^{T} \alpha_i + \sum_{i=T-k}^{T-1} \alpha_i + \cdots + \sum_{i=1}^{k} \alpha_i\right) \\
&+ \frac{2L}{nk}\left(\sum_{i=1}^{k-1} \alpha_i + \sum_{i=1}^{k-2} \alpha_i + \cdots + \sum_{i=1}^{1} \alpha_i\right).
\end{aligned}
\tag{28}
$$

Let $\alpha_{i,j} = \alpha$, we get

$$
\mathbb{E}\left[\bar{\delta}_T\right] = \frac{2\alpha L}{n}\left(T - \frac{k}{2}\right).
\tag{29}
$$

Plugging this back into Eq. 4.4 and combining the above and Lemma 4.4, we obtain

$$
\epsilon_{gen} = \mathbb{E}|F(\bar{w}_T^K; z) - F(\bar{w}_T^{K'}; z)| \leq \frac{2\alpha L^2 \hat{s}}{n}\left(T - \frac{k}{2}\right).
\tag{30}
$$

And we finish the proof.

In fact, based on the above proof, the generalization bound can be readily extended to the case of a decaying learning rate. However, we adopt a constant learning rate mainly for ease of comparison with other methods. As discussed in the remark 4.14, our approach to establishing the generalization bound of SeWA is similar to that of the paper Wang et al. (2024b), but with a fundamental difference. Our focus lies in the effect of selection on generalization, while the generalization bound of the averaged last $k$ iterates serves only as a component of our study, where a uniform weighting scheme suffices. In contrast, existing work concentrates on the paradigm of weighted averaging.

### F.3 PROOF. THEOREM 4.11 (BASED ON THE CONSTANT LEARNING RATE)

#### F.3.1 LEMMA F.1 AND IT'S PROOF

**Lemma F.1.** *Assume that $F$ is $\beta$-smooth and non-convex. Let $\alpha = \frac{c}{t}$, we have*

$$
\|w'_T - w_T\| \leq e^{\frac{c\beta k}{T-k}} \bar{\delta}_T,
\tag{31}
$$

where $\bar{\delta}_T = \frac{1}{k} \sum_{i=T-k+1}^{T} \|w'_i - w_i\|$.

**proof Lemma F.1.** By the triangle inequality and our assumption that $F$ satisfies, we have

$$\|w'_T - w_T\| = \frac{1}{k} \cdot k \cdot \|w'_T - w_T\|$$

$$\leq \frac{1}{k}(\|w'_T - w_T\| + (1 + \alpha_{T-1}\beta)\|w'_{T-1} - w_{T-1}\| + \cdots +$$

$$(1 + \alpha_{T-1}\beta)(1 + \alpha_{T-2}\beta) \cdots (1 + \alpha_{T-k+1}\beta)\|w'_{T-k+1} - w_{T-k+1}\|)$$

$$\leq \prod_{t=T-k+1}^{T} (1 + \alpha_t\beta) \left( \frac{1}{k} \sum_{i=T-k+1}^{T} \|w'_i - w_i\| \right). \tag{32}$$

Let $\alpha_t = \frac{c}{t}$, we have

$$\|w'_T - w_T\| \leq \prod_{t=T-k+1}^{T} (1 + \alpha_t\beta)\bar{\delta}_T \leq \left( 1 + \frac{c\beta}{T-k} \right)^k \bar{\delta}_T \leq e^{\frac{c\beta k}{T-k}} \bar{\delta}_T. \tag{33}$$

### F.3.2 PROOF. THEOREM 4.11

In the non-convex setting, we build the SeWA's generalization bound based on the Lemma 4.10.

Then, the last $k$ points of the averaging algorithm's stability bounds are provided as follows. Based on the relationship between $\hat{w}_T^k$ and $\hat{w}_{T-1}^k$ in Eq. 23. We consider that the different samples $z_T$ and $z'_T$ are selected to update with probability $\frac{1}{n}$ at step T.

$$\bar{\delta}_T = \bar{\delta}_{T-1} + \frac{1}{k} \sum_{i=T-k+1}^{T} \alpha_i \|\nabla F(w'_{i-1}, z_i) - \nabla F(w_{i-1}, z_i)\|$$

$$\leq \bar{\delta}_{T-1} + \frac{2\alpha_T L}{k} + \frac{1}{k} \sum_{i=T-k+1}^{T-1} \alpha_i \|\nabla F(w'_{i-1}, z_i) - \nabla F(w_{i-1}, z_i)\|, \tag{34}$$

Next, the same sample $z = z'$ is selected to update with probability $1 - \frac{1}{n}$ at step T.

$$\bar{\delta}_T \leq \bar{\delta}_{T-1} + \frac{1}{k} \sum_{i=T-k+1}^{T} \alpha_i \|\nabla F(w'_{i-1}, z_i) - \nabla F(w_{i-1}, z_i)\|$$

$$\leq \bar{\delta}_{T-1} + \frac{\alpha_T\beta}{k} \|w'_{T-1} - w_{T-1}\| + \frac{1}{k} \sum_{i=T-k+1}^{T-1} \alpha_i \|\nabla F(w'_{i-1}, z_i) - \nabla F(w_{i-1}, z_i)\| \tag{35}$$

$$\leq (1 + \frac{\alpha_T\beta e^{\frac{c\beta k}{T-k}}}{k})\bar{\delta}_{T-1} + \frac{1}{k} \sum_{i=T-k+1}^{T-1} \alpha_i \|\nabla F(w'_{i-1}, z_i) - \nabla F(w_{i-1}, z_i)\|,$$

where the proof follows from the $\beta$-smooth and Lemma F.1. Then, we bound the $\alpha\|\nabla F(w'_{T-2}, z_{T-1}) - \nabla F(w_{T-2}, z_{T-1})\|$ with different sampling.

$$\alpha_i \mathbb{E}\|\nabla F(w'_i, z_{i+1}) - \nabla F(w_i, z_{i+1})\| = \frac{2\alpha_i L}{n} + \left( 1 - \frac{1}{n} \right) \alpha_i\beta \|w_i - w'_i\|$$

$$\leq \frac{2\alpha_i L}{n} + \alpha_i\beta \left( \|w_{i-1} - w'_{i-1}\| + \alpha_{i-1}\|\nabla F(w'_{i-1}, z_i) - \nabla F'(w_{i-1}, z_i)\| \right)$$

$$\leq \frac{2\alpha_i L}{n} + \alpha_i\beta \left( \frac{2\alpha_{i-1} L}{n} + (1 + \alpha_{i-1}\beta)\|w_{i-1} - w'_{i-1}\| \right) \tag{36}$$

$$\cdots$$

$$\leq \frac{2\alpha_i L}{n} \left( 1 + \alpha_{i-1}\beta + \sum_{m=t_0}^{i-1} \prod_{t=m+1}^{i} (1 + \alpha_t\beta)\alpha_m \right) + \alpha_i\beta \prod_{t=t_0}^{i} (1 + \alpha_t\beta)\|w_{t_0} - w'_{t_0}\|,$$

where $w_{t_0} = w'_{t_0}$. Therefore, we discuss the bound for $\frac{1}{k}\sum_{i=T-k+1}^{T-1}\alpha_i\|\nabla F(w'_{i-1}, z_i) - \nabla F(w_{i-1}, z_i)\|$ based on the recursive relationship.

$$
\begin{aligned}
&\frac{1}{k}\sum_{i=T-k+1}^{T-1}\alpha_i\mathbb{E}\|\nabla F(w'_{i-1}, z_i) - \nabla F(w_{i-1}, z_i)\| \\
&\leq \frac{1}{k}\sum_{i=T-k+1}^{T-1}\frac{2\alpha_i L}{n}\left(1 + \alpha_{i-1}\beta + \sum_{m=t_0}^{i-1}\prod_{t=m+1}^{i}(1+\alpha_t\beta)\alpha_m\right) \\
&= \frac{2L}{k}\sum_{i=T-k+1}^{T-1}\frac{\alpha_i}{n} + \frac{2\beta L}{k}\sum_{i=T-k+1}^{T-1}\frac{\alpha_i\alpha_{i-1}}{n} + \frac{2L}{k}\sum_{i=T-k+1}^{T-1}\frac{\alpha_i}{n}\sum_{m=t_0}^{i-1}\prod_{t=m+1}^{i}(1+\alpha_t\beta)\alpha_m \\
&\leq \frac{2cL}{n(T-k+1)} + \frac{2\beta c^2 L}{n(T-K)^2} + \frac{2cLT^{c\beta}}{n\beta t_0^{c\beta}(T-k+1)},
\end{aligned}
\tag{37}
$$

where $\alpha_i = \frac{c}{i}$ and the proof of the last term in the first equality is provided as follows

$$
\begin{aligned}
\sum_{m=t_0}^{T-1}\prod_{t=m}^{T-1}(1+\frac{c\beta}{t})\frac{1}{m} &\leq \sum_{m=t_0}^{T-1}\frac{1}{m}\left(e^{\sum_{t=m}^{T-1}\frac{c\beta}{t}}\right) \leq \sum_{m=t_0}^{T-1}\frac{T^{c\beta}}{m^{1+c\beta}} \leq T^{c\beta}\int_{t_0}^{T-1}m^{-(1+c\beta)}dm \\
&= \frac{T^{c\beta}}{c\beta}\left(\frac{1}{t_0^{c\beta}} - \frac{1}{(T-1)^{c\beta}}\right) \leq \frac{1}{c\beta}\cdot\left(\frac{T}{t_0}\right)^{c\beta}.
\end{aligned}
\tag{38}
$$

Taking $M_1 = \left(1 + c\beta + \frac{1}{\beta}\right)$, we can obtain the bound in the expectation sense.

$$
\frac{1}{k}\sum_{i=T-k+1}^{T-1}\alpha_i\mathbb{E}\|\nabla F(w'_{i-1}, z_i) - \nabla F(w_{i-1}, z_i)\| \leq \frac{2cLM_1}{nt_0^{c\beta}}\cdot\left(\frac{1}{T-k}\right)^{1-c\beta}.
\tag{39}
$$

Compared with the results in paper Wang et al. (2024b), here we establish an upper bound on the cumulative gradient that depends on $t_0$. This enables us to derive a generalization bound that surpasses the performance of SGD in the subsequent analysis, without requiring strict assumptions.

Then, we obtain the expectation considering the above analysis

$$
\mathbb{E}\left[\bar{\delta}_{T+1}\right] \leq (1 - \frac{1}{n})\left(1 + \frac{\alpha_T\beta e^{\frac{c\beta k}{T-k}}}{k}\right)\bar{\delta}_T + \frac{1}{n}\left(\bar{\delta}_T + \frac{2\alpha_T L}{k}\right) + \frac{2cLM_1}{nt_0^{c\beta}}\cdot\left(\frac{1}{T-k}\right)^{1-c\beta}
\tag{40}
$$

let $\alpha_t = \frac{c}{t}$, then

$$
\begin{aligned}
&\leq \left(1 + (1-\frac{1}{n})\frac{c\beta e^{\frac{c\beta k}{t-k}}}{kt}\right)\bar{\delta}_t + \frac{2cL(1+kM_1)}{nk(t_0-k)^{c\beta}}\left(\frac{1}{t-k}\right)^{1-c\beta} \\
&\leq \exp\left((1-\frac{1}{n})\frac{c\beta}{kt}\right)\bar{\delta}_t + \frac{2cLM}{nk(t_0-k)^{c\beta}}\left(\frac{1}{t-k}\right)^{1-c\beta},
\end{aligned}
\tag{41}
$$

where $M = 1 + kM_1$, $c\beta \in (0, 1)$, $k < t_0$ and we used that $\lim_{x\to\infty}(1+\frac{1}{x})^x = e$ and $\lim_{x\to\infty}e^{\frac{1}{x}} = 1$.

Using the fact that $\bar{\delta}_{t_0} = 0$, we can unwind this recurrence relation from $T$ down to $t_0 + 1$.

$$
\begin{aligned}
\mathbb{E}\bar{\delta}_{t+1} &\leq \sum_{t=t_0+1}^{T} \left( \prod_{m=t+1}^{T} \exp\left( (1-\frac{1}{n})\frac{c\beta}{km} \right) \right) \frac{2cLM}{nk(t_0-k)^{c\beta}} \cdot \left( \frac{1}{t-k} \right)^{1-c\beta} \\
&= \sum_{t=t_0+1}^{T} \exp\left( \frac{(1-\frac{1}{n})c\beta}{k} \sum_{m=t+1}^{T} \frac{1}{m} \right) \frac{2cLM}{nk(t_0-k)^{c\beta}} \cdot \left( \frac{1}{t-k} \right)^{1-c\beta} \\
&\leq \sum_{t=t_0+1}^{T} \exp\left( \frac{(1-\frac{1}{n})c\beta}{k} \cdot \log(\frac{T}{t}) \right) \frac{2cLM}{nk(t_0-k)^{c\beta}} \cdot \left( \frac{1}{t-k} \right)^{1-c\beta} \\
&\leq T^{\frac{(1-\frac{1}{n})c\beta}{k}} \cdot \sum_{t=t_0+1}^{T} \left( \frac{1}{t-k} \right)^{\frac{(1-\frac{1}{n})c\beta}{k}+1-c\beta} \cdot \frac{2cLM}{nk(t_0-k)^{c\beta}} \\
&\leq \left( \frac{c\beta}{k} + 1 - c\beta \right)^{-1} \cdot \frac{2cLM}{nk(t_0-k)^{c\beta}} \cdot T^{\frac{c\beta}{k}} \cdot \left( \frac{1}{t_0-k} \right)^{\frac{c\beta}{k}-c\beta} \\
&\leq \frac{2cLM\tau}{n-1} \cdot T^{\frac{c\beta}{k}} \cdot \left( \frac{1}{t_0-k} \right)^{\frac{c\beta}{k}},
\end{aligned}
\tag{42}
$$

where $\tau = \frac{1}{k+c\beta-kc\beta}$ and $c\beta \in (0,1)$. Plugging this back into Eq. 11, we obtain

$$
\mathbb{E}|F(\bar{w}_T^k; z) - F(\bar{w}_T'^k; z)| \leq \frac{t_0}{n} + \frac{2\hat{s}cL^2M\tau}{n-1} \cdot T^{\frac{c\beta}{k}} \cdot \left( \frac{1}{t_0-k} \right)^{\frac{c\beta}{k}}.
\tag{43}
$$

By taking the extremum, we obtain the minimum

$$
t_0 = \left( \frac{2\hat{s}c^2L^2\beta M\tau}{k} \right)^{\frac{k}{k+c\beta}} \cdot T^{\frac{c\beta}{k+c\beta}} + k.
\tag{44}
$$

Finally, this setting gets

$$
\epsilon_{gen} = \mathbb{E}|F(\bar{w}_T^k; z) - F(\bar{w}_T'^k; z)| \leq \frac{1+\frac{k}{c\beta}}{n-1} \left( 2\hat{s}c^2L^2\beta\tau Mk^{-1} \right)^{\frac{k}{c\beta+k}} \cdot T^{\frac{c\beta}{c\beta+k}} + \frac{k}{n-1}.
\tag{45}
$$

To simplify, omitting constant factors that depend on $\beta$, $c$ and $L$, this setting get

$$
\epsilon_{stab} \leq \mathcal{O}_{\hat{s}}\left( \frac{T^{\frac{c\beta}{k+c\beta}}}{n} \right).
\tag{46}
$$

And we finish the proof.

