# OpenReview forum: "SeWA: Selective Weight Average via Probabilistic Masking"
_ICLR.cc/2026/Conference — Submitted to ICLR 2026_

### Official Review · Reviewer_XUU5 · 2025-10-29

**Soundness:** 3
**Presentation:** 2
**Contribution:** 2
**Rating:** 4
**Confidence:** 1

**Summary:**

Weight averaging is known to stabilize the training and gives smoother weights. This has reportedly improved model generalization.
This paper proposes selective weight averaging to minimize the bias introduced by existing state-of-the-art methods that rely on manually designed checkpoint selection rules for stochastic weight averaging and latent weight averaging.
The idea is to adaptive selects checkpoints during the final stage of training for averaging.

**Strengths:**

Strengths
- Targetting a difficult problem
- Avoids bias introduced due to manually selecting checkpoints
- Reduces the need for extensive hyper parameter tuning
- Mitigates performance degradation caused by redundant weight selection
- Formulated adaptive selection of checkpoints as a discrete subset selection problem by transforming the problem into a continuous probabilistic optimization framework.
- Derive stability-based generalization bounds which are sharper than SGD in both convex and non-convex optimization

**Weaknesses:**

Oveall, I find the idea quite interesting. I have few comments:

- I see the idea behind Figure 1, but it did not help me  understand the fact that SeWA reaches a flatter curvature (stable minima)
- Improvements reported in Figure 3 & 4 are showing marginal improvements even compared to Random

The experimentation overall seems not super convincing. But maybe theoretical guarantees claimed in the paper might be more promising. Unfortunately, this work is not in my expertise which is why I could not closely check the mathematical details.

I would need to rely on the fellow reviewers to verify the math. Experimentally, I am not fully convinced of the benefits of the proposed approach.

**Questions:**

- The difference between SeWA vs SWA is not clear in the introduction. If my understanding is correct, SWA is using previous K  epochs as well. How is SeWA different in that aspect?
- Is the analysis still valid if a model is fine-tuned after being trained with SeWA?

---

> ### Author Response · Authors · 2025-11-21
> **Response to Reviewer XUU5**
>
> **Response to Weakness 1**:
>
> Figure 1 presents a comparison of the convergence performance of algorithms employing different averaging strategies. While the SeWA algorithm still approaches a local optimum through an averaging mechanism, the key emphasis in Figure 1 is that, compared to other methods, SeWA adaptively selects a limited number of checkpoints along the training trajectory through a learnable process. This differs from the manually designed checkpoint collection frameworks used in other algorithms. Our approach requires no prior knowledge and can serve as a theoretically grounded, plug-and-play solution applicable to a wide range of tasks.
>
> The conclusion that the SeWA algorithm converges to a critical point with flatter curvature is primarily guaranteed by the theorem 4.2.
>
> **Response to Weakness 2**:
>
> Thank you for pointing this out. I'd be happy to explore the underlying reasons behind the experimental phenomena with you.
>
> **A:** This is not a weakness specific to the SeWA algorithm in such tasks, but rather a common characteristic shared by all algorithms, as demonstrated by our supplementary experiments.
>
> We conduct experiments on CIFAR100 using VGG16 network, and compare with SGD, SWA, LAWA, and EMA. The SGD achives accuracy 72.9%. In the following table, we present the accuracy improvements of each method over SGD. It shows that our method achieves considerable gain over other averaging methods.
>
> | Method | K = 10 | K = 20 | K = 30 | K = 40 | K = 50 |
> |--------|--------|--------|--------|--------|--------|
> | SWA    | -0.05  | +0.03  | +0.09  | +0.12  | +0.06  |
> | LAWA   | -0.01  | +0.01  | +0.01  | +0.05  | +0.04  |
> | EMA    | -0.10  | +0.08  | +0.06  | 0.00   | -0.20  |
> | SeWA   | +0.26  | +0.26  | +0.41  | +0.51  | +0.49  |
>
> As shown in the above table, our method shows considerable gain over other averaging methods including SWA and EMA. In the image classification task, the training dyanmics are very stable, therefore we think that averaging methods are not so significant here. For example, SWA, LAWA, and EMA do not improve the performance much.
>
> **B:** For tasks like image and text classification, where the training process is relatively stable, most models generated by updates in the later stages of training already incorporate sufficient training information and exhibit converging performance. This limits the extent to which averaging algorithms can further enhance the final model's performance.
>
> We conducted extended experiments on the NYUv2 dataset using the ViT model, achieving average improvements of +4.89% on the training set and +1.88% on the validation set.
>
> | Method | 2300 | 4300 | 6300 | 8300 (iterations)|
> |--------|--------|--------|--------|--------|
> | Baseline(Train)| 0.1668 | 0.1343 | 0.1237 | 0.1235  |
> | SeWA  (Train) | 0.1463 | 0.1324 | 0.1207  | 0.1160  |
> | Baseline(Val)| 0.1970 | 0.1719 | 0.1683 | 0.1680  |
> | SeWA  (Val) | 0.1813 | 0.1689 | 0.1674  | 0.1648  |
>
> Due to the relatively stable training process of the task, the SeWA algorithm showed the same improvement effect when applied to additional regression problems. However, in the behaviour cloning experiments (Section 5.1), where the training is more unstable, our method shows more significant improvements.
>
> **C:** These results align with our theoretical expectations for the algorithm and further demonstrate SeWA's applicability to a broader range of environments. The experimental results serve as validation for our theoretical exploration, but the SeWA algorithm itself and its underlying operational principles should be the primary focus of attention.
>
> **Response to Question 1**:
>
> Both SWA and SeWA share the same research rationale for improving generalization: they guide the final model into a flat region of low loss by averaging parameters in the later stages of training, thereby achieving significant generalization performance. However, their implementation strategies differ considerably—particularly in how they collect parameters for averaging.
>
> SWA requires a decaying learning rate and collects the checkpoint with the lowest loss at the end of each training cycle. It then performs an equally weighted average over all collected points. In contrast, SeWA does not change the original training trajectory. Instead, it learns which saved checkpoints should be included in the averaging process.
>
> **Response to Question 2**:
>
> First, the theoretical explanation that SeWA itself converges to a critical point with flatter curvature of the loss landscape remains valid.
>
> Second, the stability theory studies how sensitive an algorithm's output is to input perturbations. This is critical here because performing fine-tuning after SeWA causes the final output to depend heavily on the specific fine-tuning settings, thus necessitating a deeper investigation into its stability.

---

### Official Review · Reviewer_yKEf · 2025-10-30

**Soundness:** 2
**Presentation:** 2
**Contribution:** 2
**Rating:** 4
**Confidence:** 3

**Summary:**

The paper proposes the SeWA algorithm, which is a post-training checkpoint averaging method that learns which $K>0$ of the last $k$ checkpoints to include in the final average. It relaxes the discrete selection with Gumbel-Softmax and optimizes it using Monte Carlo (iteratively). The paper provides theoretical results for evaluating generalization error in both convex and non-convex settings. It also provides empirical results over diverse domains, including behavior cloning (D4RL), CIFAR-100, and AG News.

**Strengths:**

* The paper provides tighter theoretical results for convex and non-convex settings (the proofs were not checked).
* The paper uses standard assumptions of smoothness and Lipschitz.
* The experiments were tested on diverse domains: image, text, and locomotion trajectories.

**Weaknesses:**

* In practice, if you select the final model on validation at multiple checkpoints, you would need to run SeWA at each validation point (or at least repeatedly near the end), which can be expensive compared to the other approaches that do not require this procedure and can be used directly. The paper would benefit from reporting wall-clock time per SeWA run, relative to one training epoch, for the chosen $K$, $k$, $M$, and max_iterations in each experiment. Also, the paper should specify the number of SeWA optimization iterations (max_iteration) and the value of $M$ used per experiment. Currently, only a D4RL ablation in the appendix touches on this, but it's still hard to capture the full cost of the proposed approach.
* The paper also should discuss the storage overhead of additional $k$ checkpoints versus the other approaches, which keep a single running aggregate
* In Figures 3 and 4, the results for SeWA and LAWA appear identical across different values of $K$. Also, SGD takes longer to converge on CIFAR-100; the authors may consider more complex architectures better suited to this task. For example, in [1] (TWA), they used ResNet and VGG and achieved more stable convergence and better accuracy on TWA than SGD (around 80%).
* The results in Figures 3 and 4 do not show a significant improvement. Given the costly SWeA optimization process, averaging $k$ checkpoints at random yields results similar to those from the complete optimization and requires neither extensive hyperparameter tuning nor a costly optimization procedure. Also, EMA and SWA should be added to these CIFAR-100/AG News experiments, not only D4RL.
* The authors did not provide any supplementary material. Since no code for the empirical evaluation is available, the community cannot verify or assess the proposed approach's contribution.


[1] Li, Tao, et al. "Trainable weight averaging: Efficient training by optimizing historical solutions." The Eleventh International Conference on Learning Representations. 2022.

**Questions:**

1. Regarding the experiments in Figure 2, wouldn’t a fairer comparison be to average the SWA with $K$ checkpoints over the last $k$ iterations, rather than every $K$ iterations? The same for EMA: a fairer comparison is to use a weight decay of $1-K$, which effectively averages the last $K$ checkpoints.
2. Did the authors try a trainable approach as done with TWA [1]? Why not add TWA to the comparison?

[1] Li, Tao, et al. "Trainable weight averaging: Efficient training by optimizing historical solutions." The Eleventh International Conference on Learning Representations. 2022.

---

> ### Author Response · Authors · 2025-11-21
>
> **Response to Weakness 1**:
>
> 1. Thanks for raising this meaningful question. Indeed, once the pretraining process converges, we begin collecting checkpoints to construct a selected pool, and subsequently, we only need to run SeWA once to complete the entire workflow. Unlike SWA or LAWA, which emphasize human-designed collection strategies, SeWA leverages a learnable process to mitigate the impact of the collection strategy.
>
>
> 2. We acknowledge that computational cost is a widespread concern in this field, and we will add relevant discussions in the revision. To investigate this aspect, we conduct additional ablations by increasing the number of MC samples $M$.
>
> - First, we fix $K = 50$ (number of selected checkpoints) and report results on two D4RL tasks, increasing $M$ from 1 to 20. Results are shown in the following table. (A) Increasing $M$ significantly improves the performance due to more accurate gradient estimation. (B) The computional cost is acceptable; using 20x times particles only requirs about 1.44x times wall-clock time. According to these experiments, while our method already shows advantages over other baselines, increasing the sampling number further improves the performances at relatively slight computational costs. In practice, we can always choose the sampling number based on our computational budget.
>
> | $M$ | 1  | 5 | 10 | 20 |
> | -------- | -------- | -------- | - | -|
> |wall-clock time | 0.25 s/iter | 0.31 s/iter | 0.32 s/iter | 0.36 s/iter |
> | hopper-medium     | 1563.24 ± 46.98   |  1568.38 ± 37.06  | 1575.96 $\pm$ 20.95 | 1593.62 ± 8.86 |
> | hopper-medium-expert |1616.12 ± 10.58  | 1620.11 ± 9.33 | 1625.15 ± 7.95| 1642.37 ± 4.12|
>
> 3. In all of our experiments, we set the number of mask optimization steps (max_iteration) to 10% of the total pretraining steps (e.g., 1000 out of 10,000). This choice was made to ensure that the additional computational cost remains bounded and practical, while still allowing the optimization to converge to a high-quality selection mask. This schedule was determined empirically based on ablation studies, which showed that performance plateaus well before 1000 steps, and increasing iterations beyond this point provided diminishing returns
>
> We measured the average per-iteration wall-clock time for both the pretraining and the mask-learning phases to assess their relative cost. Although a single mask optimization step is more expensive—primarily due to:  averaging $K$ checkpoints at each iteration, and Performing a forward pass over the validation set. the total number of optimization steps is much smaller (e.g., 10000 vs. 100,000), keeping the overall cost marginal. Total overhead: For typical D4RL tasks, SeWA adds <25% to the total wall-clock time, making it a lightweight extension in practice.
>
> | pretraining | mask optimization |
> | -------- | -------- |
> | 0.10 s/iter   | 0.22 s/iter   |
>
> **Response to Weakness 2**:
>
> The addtional storage overhead is the $k$ times size of checkpoints (330Kb for ResNet1, 245K for Decision Transformer). We will add this discussion in the manuscript.
>
> In fact, when the number of fixed average checkpoints is fixed, the additional storage overhead is the same for offline averaging algorithms such as SWA and LAWA. Although EMA does not incur this cost, it must perform a weighted average at each step, which incurs additional computational overhead.

---

> ### Author Response · Authors · 2025-11-24
>
> **Response to Weakness 3 & 4**:
>
> Thank you for pointing this out. I'd be happy to explore the underlying reasons behind the experimental phenomena with you.
>
> **A:** This is not a weakness specific to the SeWA algorithm in such tasks, but rather a common characteristic shared by all algorithms, as demonstrated by our supplementary experiments.
>
> We conduct experiments on CIFAR100 using VGG16 network, and compare with SGD, SWA, LAWA, and EMA. The SGD achives accuracy 72.9%. In the following table, we present the accuracy improvements of each method over SGD. It shows that our method achieves considerable gain over other averaging methods.
>
> | Method | K = 10 | K = 20 | K = 30 | K = 40 | K = 50 |
> |--------|--------|--------|--------|--------|--------|
> | SWA    | -0.05  | +0.03  | +0.09  | +0.12  | +0.06  |
> | LAWA   | -0.01  | +0.01  | +0.01  | +0.05  | +0.04  |
> | EMA    | -0.10  | +0.08  | +0.06  | 0.00   | -0.20  |
> | SeWA   | +0.26  | +0.26  | +0.41  | +0.51  | +0.49  |
>
> As shown in the above table, our method shows considerable gain over other averaging methods including SWA and EMA. In the image classification task, the training dyanmics are very stable, therefore we think that averaging methods are not so significant here. For example, SWA, LAWA, and EMA do not improve the performance much.
>
> **B:** For tasks like image and text classification, where the training process is relatively stable, most models generated by updates in the later stages of training already incorporate sufficient training information and exhibit converging performance. This limits the extent to which averaging algorithms can further enhance the final model's performance.
>
> We conducted extended experiments on the NYUv2 dataset using the ViT model, achieving average improvements of +4.89% on the training set and +1.88% on the validation set.
>
> | Method | 2300 | 4300 | 6300 | 8300 (iterations)|
> |--------|--------|--------|--------|--------|
> | Baseline(Train)| 0.1668 | 0.1343 | 0.1237 | 0.1235  |
> | SeWA  (Train) | 0.1463 | 0.1324 | 0.1207  | 0.1160  |
> | Baseline(Val)| 0.1970 | 0.1719 | 0.1683 | 0.1680  |
> | SeWA  (Val) | 0.1813 | 0.1689 | 0.1674  | 0.1648  |
>
> Due to the relatively stable training process of the task, the SeWA algorithm showed the same improvement effect when applied to additional regression problems. However, in the behaviour cloning experiments (Section 5.1), where the training is more unstable, our method shows more significant improvements.
>
> **C:** These results align with our theoretical expectations for the algorithm and further demonstrate SeWA's applicability to a broader range of environments. The experimental results serve as validation for our theoretical exploration, but the SeWA algorithm itself and its underlying operational principles should be the primary focus of attention.
>
>
> **Response to Weakness 5**:
>
> The source code for the algorithm implementation and comparative experiments will be made publicly available upon the acceptance of this paper.
>
> **Response to Question 1**:
>
> For a fair comparison with SWA, as you mentioned, "averaging $K$ points over the last $k$ iterations," this setup is equivalent to LAWA in our extended experiments. We tested K=10, 20, 30, 40, and 50, and the results consistently demonstrate that the SeWA algorithm achieves a significant performance advantage.
>
> For a fair comparison with EMA, to be honest, I did not understand the "$1-K$ decay" setting, since K represents the number of averaged points. I considered it as $1-\frac{1}{K}$, but this expression actually corresponds to averaging K points with equal weights, which is equivalent to SWA. In our extended experiments above, we used a decay factor of 0.95.

---

> ### Author Response · Authors · 2025-11-27
>
> **Response to Question 2**:
>
> Now, we are also training using the TWA approach in the tail phase of the training process, which involves learning a set of continuous weight coefficients.
>
> We conduct experiments of training VGG16 network on the CIFAR-10 dataset, following the setting in the TWA paper. We apply SeWA on the 100 tail training checkpoints, learning masks for additional 10 epochs, which is the same with TWA. The number of chosen checkpoints is set to $K \in \{10, 20, 30\}$, and the MC samples is set to $M = 1$, which makes the training cost identical to TWA. The results are shown in the following table. Our method tends to outperform TWA. Due to the limited time, we do not conduct more experiments, but we would like to test more datasets and networks in the revision.
>
> | TWA | SeWA (K=10) | SeWA (K=20) | SeWA (K=30) |
> | -------------- | ----------- | ----------- | ----------- |
> | 94.16 $\pm$ 0.14 | 94.26 $\pm$ 0.02 | 94.30 $\pm$ 0.02 | 94.28 $\pm$ 0.05 |
>
> We did not compare TWA and SeWA in the submitted manuscript because we believe that a direct comparison would be unfair for the following reasons:
>
> 1. SeWA is designed as a post-training algorithm aimed at improving generalization. It intentionally ignores the parameter learning of the early learning stage and only leverages the later stage, information-saturated parameters. In contrast, TWA learns both parameters and weighting coefficients throughout the entire training process. Therefore, it is not feasible to compare them using quantitative metrics (such as the number of iterations T or the averaging window size K) on an equal footing.
>
> 2. The computational cost of SeWA primarily comes from the mask-learning process, which occurs only after the base training is complete. However, TWA performs extra orthogonalization during every parameter update, and this additional computational overhead is incurred throughout the entire training process.

---

> > ### Comment · Reviewer_yKEf · 2025-11-27
> >
> > Thank you for the rebuttal; I appreciate the clarifications. I will retain my score.

---

### Official Review · Reviewer_h3s3 · 2025-10-30

**Soundness:** 3
**Presentation:** 3
**Contribution:** 2
**Rating:** 4
**Confidence:** 4

**Summary:**

This work addresses an adaptive weight selecting method for weight averaging method of training DNNs at the training tail stage. Specifically, it formulates a discrete scheme of determining the binary variables for choosing the checkpoints, and then leverage GS estimator to continualize the optimization to fit the training framework. Then, some analytical results are presented, so are the experimental evaluations.

**Strengths:**

The continuation of the weight averaging is a good idea on the class of weight averaging methods for improving the generalization ability of DNNs.

**Weaknesses:**

1. The results are insufficient: except Table 2, the results (e.g.g, Figure 3 and 4 for image class-action and text classification) do not show distinctive improvements so  the competitiveness  lacks convincing supports; otherwise, the performance improvement look increment in general setups.  Besides, the evaluated network architectures are datasets can be extended for comprehensiveness.
2. As it claims its particular effectiveness on RL that may have more unstable training. Would it be possible to have particular theoretical analysis focusing on such task and settings?
3. Similar to the insight in the above bullet, at the early training stage “more unstable training” can be expected, what if SeWA is applied in such scenario? (TWA claims its advantage also at the early training stage, as it also somehow optimizes the weights. By the way, why not compare with TWA in the experiments?).

**Questions:**

See the weakness.

---

> ### Author Response · Authors · 2025-11-21
>
> **Response to Weakness 1**:
>
> Thank you for pointing this out. I'd be happy to explore the underlying reasons behind the experimental phenomena with you.
>
> **A:** This is not a weakness specific to the SeWA algorithm in such tasks, but rather a common characteristic shared by all algorithms, as demonstrated by our supplementary experiments.
>
> We conduct experiments on CIFAR100 using VGG16 network, and compare with SGD, SWA, LAWA, and EMA. The SGD achives accuracy 72.9%. In the following table, we present the accuracy improvements of each method over SGD. It shows that our method achieves considerable gain over other averaging methods.
>
> | Method | K = 10 | K = 20 | K = 30 | K = 40 | K = 50 |
> |--------|--------|--------|--------|--------|--------|
> | SWA    | -0.05  | +0.03  | +0.09  | +0.12  | +0.06  |
> | LAWA   | -0.01  | +0.01  | +0.01  | +0.05  | +0.04  |
> | EMA    | -0.10  | +0.08  | +0.06  | 0.00   | -0.20  |
> | SeWA   | +0.26  | +0.26  | +0.41  | +0.51  | +0.49  |
>
> As shown in the above table, our method shows considerable gain over other averaging methods including SWA and EMA. In the image classification task, the training dyanmics are very stable, therefore we think that averaging methods are not so significant here. For example, SWA, LAWA, and EMA do not improve the performance much.
>
> **B:** For tasks like image and text classification, where the training process is relatively stable, most models generated by updates in the later stages of training already incorporate sufficient training information and exhibit converging performance. This limits the extent to which averaging algorithms can further enhance the final model's performance.
>
> We conducted extended experiments on the NYUv2 dataset using the ViT model, achieving average improvements of +4.89% on the training set and +1.88% on the validation set.
>
> | Method | 2300 | 4300 | 6300 | 8300 (iterations)|
> |--------|--------|--------|--------|--------|
> | Baseline(Train)| 0.1668 | 0.1343 | 0.1237 | 0.1235  |
> | SeWA  (Train) | 0.1463 | 0.1324 | 0.1207  | 0.1160  |
> | Baseline(Val)| 0.1970 | 0.1719 | 0.1683 | 0.1680  |
> | SeWA  (Val) | 0.1813 | 0.1689 | 0.1674  | 0.1648  |
>
> Due to the relatively stable training process of the task, the SeWA algorithm showed the same improvement effect when applied to additional regression problems. However, in the behaviour cloning experiments (Section 5.1), where the training is more unstable, our method shows more significant improvements.
>
> **C:** These results align with our theoretical expectations for the algorithm and further demonstrate SeWA's applicability to a broader range of environments. The experimental results serve as validation for our theoretical exploration, but the SeWA algorithm itself and its underlying operational principles should be the primary focus of attention.
> **Response to Weakness 2**:
>
> In this paper, Theorem 4.2 demonstrates that SeWA converges to a critical point with flatter curvature, providing a theoretical explanation for its improved generalization. Theorems 4.6 and 4.11 further establish, through stability analysis, that SeWA enjoys a tighter generalization bound, offering theoretical guarantees for its performance gains. Moreover, we position SeWA as a general-purpose plugin that enhances final model generalization in a post-training manner, as it operates solely on checkpoints along the training trajectory and is independent of any specific task. In this sense, task-specific theoretical analysis appears unnecessary.
>
> Furthermore, this paper emphasizes that SeWA brings more pronounced improvements in less stable training settings such as reinforcement learning (RL), for two main reasons:
> 1. Averaging is a core SeWA mechanism. It is well established that averaging naturally reduces variance in stochastic outputs.
> 2. Tasks with high loss fluctuation, such as RL, benefit more significantly from averaging. In contrast, tasks with homogeneous model performance see diminishing returns from averaging.

---

> ### Author Response · Authors · 2025-11-24
>
> **Response to Weakness 3**:
>
> In fact, we have experimentally demonstrated the effectiveness of SeWA even in the early stages of training. As shown in Figures 3 and 4, the red curve corresponding to SeWA consistently lies above those of other algorithms. This indicates that SeWA consistently identifies locally critical points that enhance performance, even when training is still under-equipped in its early stage. However, this advantage comes at the cost of additional computation and storage.
>
> We did not compare TWA and SeWA in the submitted manuscript because we believe that a direct comparison would be unfair for the following reasons:
>
> 1. SeWA is designed as a post-training algorithm aimed at improving generalization. It intentionally ignores the parameter learning of the early learning stage and only leverages the later stage, information-saturated parameters. In contrast, TWA learns both parameters and weighting coefficients throughout the entire training process. Therefore, it is not feasible to compare them using quantitative metrics (such as the number of iterations T or the averaging window size K) on an equal footing.
>
> 2. The computational cost of SeWA primarily comes from the mask-learning process, which occurs only after the base training is complete. However, TWA performs extra orthogonalization during every parameter update, and this additional computational overhead is incurred throughout the entire training process.
>
> Now, we are also training using the TWA approach in the tail phase of the training process, which involves learning a set of continuous weight coefficients. We conduct experiments of training VGG16 network on the CIFAR-10 dataset, following the setting in the TWA paper. We apply SeWA on the 100 tail training checkpoints, learning masks for additional 10 epochs, which is the same with TWA. The number of chosen checkpoints is set to $K \in \{10, 20, 30 \}$, and the MC samples is set to $M = 1$, which makes the training cost identical to TWA. The results are shown in the following table. Our method tends to outperform TWA. Due to the limited time, we do not conduct more experiments, but we would like to test more datasets and networks in the revision.
>
> | TWA | SeWA (K=10) | SeWA (K=20) | SeWA (K=30) |
> | -------------- | ----------- | ----------- | ----------- |
> | 94.16 $\pm$ 0.14 | 94.26 $\pm$ 0.02 | 94.30 $\pm$ 0.02 | 94.28 $\pm$ 0.05 |

---

### Official Review · Reviewer_VxcC · 2025-10-31

**Soundness:** 3
**Presentation:** 3
**Contribution:** 2
**Rating:** 4
**Confidence:** 4

**Summary:**

This paper introduces SeWA, Selective Weight Averaging. Instead of averaging the last k checkpoints by rule, SeWA learns which checkpoints to average by casting selection as a probabilistic masking problem.
In addition, this paper uses large text space for theory analysis.
SeWA is evaluated on D4RL behavior cloning, CIFAR-100, and AG News.

**Strengths:**

1. This paper is well motivated and has strong theoretical backing. The paper provides both flatness convergence arguments and stability-based generalization bounds, comparing favorably with SGD, SWA, and LAWA under convex and non-convex assumptions.
2. The experiments span three domains (RL, vision, and text), showing consistent improvements across varied architectures and data distributions.

**Weaknesses:**

1. while the paper emphasizes SeWA’s simplicity, its sample-based optimization introduces additional forward passes and probability updates that are not accounted for in the baseline comparisons. A fair comparison should including additional computational consumption on these.
2. Except for the RL experiments, most performance curves show nearly overlapping trajectories in the figures. The visual and quantitative margins are subtle in the curve plot only form.
3. The sample-based optimization uses the training objective F as the criterion (as stated in sec. 2) for selecting checkpoints. This implicitly assumes that the training and test loss landscapes are sufficiently aligned, which rarely holds in realistic non-convex regimes. Consequently, SeWA might overfit to training dynamics when the learned selection probabilities are tuned purely on training losses.
4. Since SeWA relies on sampling and averaging across multiple candidate checkpoints, it requires storing a substantial portion (or even all) of the recent checkpoints in storage consumption. This can become expensive for large-scale or long-horizon training (e.g., vision transformers, LLMs), where checkpoints are heavy and I/O-bound.

**Questions:**

All my concerns/questions are listed in the weakness section.

---

> ### Author Response · Authors · 2025-11-21
> **response to the Reviewer VxcC**
>
> **Response to Weakness 1**:
>
> Unlike traditional methods that require costly hyperparameter sweeps (e.g., selecting which models to average), SeWA introduces a lightweight post-hoc mask optimization phase (Algorithm 1). This phase adds only a small computational overhead—typically 10% of pretraining time (e.g., 1,000 steps vs. 10,000)—while significantly reducing manual tuning.
>
> To further address the “equal-compute” concern, we conducted an SGD-enhanced experiment where SGD was allowed to continue training for additional steps, matching SeWA’s computational cost (including the overhead from multiple evaluations of $F$ during mask optimization). Even under this extended budget, the performance of SGD-enhanced remains consistently lower than SeWA, as shown below:
>
> | Column 1 | SGD | SGD_enhanced | SeWA |
> | -------- | -------- | -------- | - |
> | Hopper-medium     | 1245.04 $\pm$ 69.32  |1275.34 $\pm$ 67.31     | 1389.28 $\pm$ 46.98     |
> | Hopper-medium-expert| 1460.79 $\pm$ 15.31 | 1480.85 $\pm$ 15.22 | 1616.12 $\pm$ 10.58 |
>
> While the best SGD checkpoint provides a strong baseline, it suffers from high variance and lacks robustness across tasks. Merely running longer does not guarantee improvement due to oscillations near local minima—a common issue in late-stage SGD. In contrast, SeWA adaptively aggregates informative checkpoints, effectively stabilizing the optimization trajectory and consistently achieving better generalization with fewer averaged points. This demonstrates that SeWA’s gains are not just from increased compute, but from a principled selection mechanism that enhances stability and generalization.
>
>
> **Response to Weakness 2**:
>
> Thank you for pointing this out. I'd be happy to explore the underlying reasons behind the experimental phenomena with you.
>
> **A:** This is not a weakness specific to the SeWA algorithm in such tasks, but rather a common characteristic shared by all algorithms, as demonstrated by our supplementary experiments.
>
> We conduct experiments on CIFAR100 using VGG16 network, and compare with SGD, SWA, LAWA, and EMA. The SGD achives accuracy 72.9%. In the following table, we present the accuracy improvements of each method over SGD. It shows that our method achieves considerable gain over other averaging methods.
>
> | Method | K = 10 | K = 20 | K = 30 | K = 40 | K = 50 |
> |--------|--------|--------|--------|--------|--------|
> | SWA    | -0.05  | +0.03  | +0.09  | +0.12  | +0.06  |
> | LAWA   | -0.01  | +0.01  | +0.01  | +0.05  | +0.04  |
> | EMA    | -0.10  | +0.08  | +0.06  | 0.00   | -0.20  |
> | SeWA   | +0.26  | +0.26  | +0.41  | +0.51  | +0.49  |
>
> As shown in the above table, our method shows considerable gain over other averaging methods including SWA and EMA. In the image classification task, the training dyanmics are very stable, therefore we think that averaging methods are not so significant here. For example, SWA, LAWA, and EMA do not improve the performance much.
>
> **B:** For tasks like image and text classification, where the training process is relatively stable, most models generated by updates in the later stages of training already incorporate sufficient training information and exhibit converging performance. This limits the extent to which averaging algorithms can further enhance the final model's performance.
>
> We conducted extended experiments on the NYUv2 dataset using the ViT model, achieving average improvements of +4.89% on the training set and +1.88% on the validation set.
>
> | Method | 2300 | 4300 | 6300 | 8300 (iterations)|
> |--------|--------|--------|--------|--------|
> | Baseline(Train)| 0.1668 | 0.1343 | 0.1237 | 0.1235  |
> | SeWA  (Train) | 0.1463 | 0.1324 | 0.1207  | 0.1160  |
> | Baseline(Val)| 0.1970 | 0.1719 | 0.1683 | 0.1680  |
> | SeWA  (Val) | 0.1813 | 0.1689 | 0.1674  | 0.1648  |
>
> Due to the relatively stable training process of the task, the SeWA algorithm showed the same improvement effect when applied to additional regression problems. However, in the behaviour cloning experiments (Section 5.1), where the training is more unstable, our method shows more significant improvements.
>
> **C:** These results align with our theoretical expectations for the algorithm and further demonstrate SeWA's applicability to a broader range of environments. The experimental results serve as validation for our theoretical exploration, but the SeWA algorithm itself and its underlying operational principles should be the primary focus of attention.

---

> ### Author Response · Authors · 2025-11-24
>
> **Response to Weakness 3**:
> 1. Training models on the training set and evaluating their performance on the test set represents the standard paradigm in machine learning. The rationale behind optimizing SeWA using training loss is grounded in this principle, as no information from the test set should be incorporated during training.
>
> 2. As rightly noted by the reviewer, "in reality, the loss landscapes of training and test data can differ." Therefore, our proposed method aims to enhance model generalization through a simple yet efficient algorithm. In this paper, we formally prove in Theorem 4.2 that SeWA converges to a critical point with flatter curvature—a property widely associated with better generalization. Furthermore, through stability analysis in Theorems 4.6 and 4.11, we demonstrate that SeWA indeed enjoys a tighter generalization error bound.
>
> **Response to Weakness 4**:
>
> When model scale becomes sufficiently large, storage consumption indeed poses an unavoidable challenge for all offline averaging algorithms. Although this inherent difficulty cannot be overcome, we thank the reviewer for their interest in this issue and look forward to further discussion on the matter.
>
> 1. Firstly, we do not advocate for saving most (or even all) checkpoints. The premise of running SeWA is during the late stage of training, when parameter performance tends to saturate. This implies that these points lie near local optima on the loss landscape, creating a foundation for performance improvement through averaging. However, introducing too many checkpoints—especially those with under-trained parameters—into the computation process can hinder algorithm convergence.
>
> 2. Secondly, as demonstrated in our D4RL experiments, averaging just 10 checkpoints yields significantly better results than other methods using 100 checkpoints. This indicates that only a select few checkpoints (or checkpoints located within specific regions) genuinely contribute to performance gains. In this context, strategically sampling checkpoints at intervals during the tail stage is an effective approach—one we have adopted in practice.
>
> 3. Additionally, quantizing the checkpoints in the pool to reduce memory costs represents a promising direction for our future research.

---

### Author Response · Authors · 2025-11-26
**Summary of Rebuttal**

We would like to thank all the reviewers for their constructive feedback and insightful suggestions. In our individual responses, we have addressed these points in detail, including:

1.  **Experiments:**
    * The performance in image (CIFAR) and text classification (AG News) appears insignificant. **(Reviewers: VxcC, h3s3, yKEf, XUU5)**
    * EMA and SWA should be added to these CIFAR-100/AG News experiments, not only D4RL. **(Reviewers: yKEf)**
    * Comparison with **TWA**, which also somehow optimizes the weights. **(Reviewers: yKEf, h3s3)**
     * SeWA introduces additional computational costs (mask optimization, forward passes) that are not accounted for in baseline comparisons. A fair comparison should use "equal compute" (e.g., allowing SGD to run longer). **(Reviewer: VxcC)**
    * The paper lacks specific reporting on wall-clock time, the number of Monte Carlo samples ($M$), and optimization iterations relative to training epochs. **(Reviewer: yKEf)**

**Our response:** We provided **new tables** for:

* **CIFAR-100 (VGG16):** Showed SeWA outperforms SGD, SWA, LAWA, and EMA significantly (e.g., +0.51% over SGD vs. SWA's +0.12%).
* **NYUv2 (Vision Transformers):** Demonstrated same improvements in regression tasks (+1.88% on validation).

**Argument:** We emphasized that it is not a weakness specific to the SeWA algorithm in such tasks, but rather a common characteristic shared by all algorithms (EMA,SWA,LAWA). We admitted that image classification is often too stable for averaging to show massive gains. SeWA shines most in **unstable** training regimes (like Behavioral Cloning/RL), where variance reduction is most needed. **(Response to Reviewer VxcC's Weakness 2)**

* **TWA Comparison:** We added a direct comparison showing SeWA slightly outperforms TWA on CIFAR-10 while being a post-hoc method (unlike TWA which adds cost throughout training).

**Argument:** We experimentally validated the superiority of SeWA over TWA and explained why we did not compare it with TWA in the manuscript. **(Response to Reviewer h3s3's Weakness 3)**

* **SGD-Enhanced Experiment:** To answer the "equal-compute" critique, we ran SGD for additional steps equivalent to the time SeWA takes for mask optimization. SeWA still outperformed this extended SGD, proving the gain comes from the method, not just more compute. **(Response to Reviewer VxcC's Weakness 1)**
* **Wall-Clock Analysis and Hyperparameter:** We provided data showing SeWA adds only a small overhead (typically **<25%** of total time, or 10% of steps) and is practically lightweight. We analyzed the impact of Monte Carlo samples ($M$) and batch size, showing $M=1$ is often sufficient, but increasing $M$ improves results with acceptable cost. **(Response to Reviewer yKEf's Weakness 1)**

2.  **Discussions & Mechanism Concerns:**
    * Optimizing based on *training* loss might lead to overfitting since the training and test landscapes differ. **(Reviewer: VxcC)**
    * The claim of "flatter curvature" is not clearly visualized or proven by Figure 1. **(Reviewer: XUU5)**
    * If the specific theoretical analysis for Reinforcement Learning (RL) settings is requested? **(Reviewer: h3s3)**
    * Saving multiple checkpoints creates a high storage overhead, which is expensive for large-scale models (LLMs, ViTs). **(Reviewers: VxcC, yKEf)**

**Our response:** We have provided detailed explanations and discussions regarding the above issues.

* **Generalization Theory:** They referred to **Theorem 4.2**, proving SeWA converges to critical points with flatter curvature, and **Theorems 4.6/4.11** regarding stability bounds. **(Response to Reviewer VxcC's Weakness 3)**
* **Training vs. Test Loss:** They argued that standard ML optimizes training loss; SeWA’s mechanism specifically selects checkpoints that stabilize the trajectory, which theoretically links to better generalization on test sets. **(Response to Reviewer XUU5's Weakness 1)**
* **RL Theory:** They clarified that SeWA is a general-purpose plugin; specific RL theory is unnecessary because the benefit comes from the general principle of averaging reducing variance in high-fluctuation environments. **(Response to Reviewer h3s3's Weakness 2)**
* **Efficiency Strategy:** They argued that one does not need to save *all* checkpoints. Experiments showed that averaging just **10 checkpoints** (tail stage) often beats methods averaging 100, minimizing storage needs. **(Response to Reviewer VxcC's Weakness 4)**

Once again, we sincerely thank area chairs and all reviewers for their detailed feedback. We will integrate all corresponding discussions, analyses, and additional experimental results into the revised manuscript.

---

### Meta-Review · Area_Chair_JqJg · 2026-01-01

**Summary:**

This paper introduces Selective Weight Averaging (SeWA), a method that adaptively selects checkpoints for averaging in the final training stages, and demonstrates its superior generalization capability both theoretically and empirically.
The reviewers found the core idea of this paper to be interesting, tackling a challenging problem. They acknowledged the mathematical rigor of the work and the contributions made, including both flatness convergence arguments and stability-based generalization bounds, which lead to tighter theoretical results. However, all reviewers unanimously agreed that the experimental validation remains insufficient.

I appreciate the authors’ comprehensive rebuttal and the additional experimental results they provided. Nevertheless, given the highly competitive nature of ICLR this year and the consistent scores from the reviewers, I am unable to accept this paper for publication.

**Reviewer Concerns:**

Almost all concerns revolve around the insufficiency of experimental results, and it appears unlikely that the additional experimental findings submitted during the rebuttal can fully address them.

**Reviewer Scores:**

Since the reviewers did not respond to the authors, their scores are unlikely to change.

---

### Decision · Program_Chairs · 2026-01-26

Reject